# A Comparative Study of the Antiemetic Effects of α_2_-Adrenergic Receptor Agonists Clonidine and Dexmedetomidine against Diverse Emetogens in the Least Shrew (*Cryptotis parva*) Model of Emesis

**DOI:** 10.3390/ijms25094603

**Published:** 2024-04-23

**Authors:** Yina Sun, Nissar A. Darmani

**Affiliations:** Department of Basic Medical Sciences, College of Osteopathic Medicine of the Pacific, Western University of Health Sciences, 309 East Second Street, Pomona, CA 91766, USA; yinasun@westernu.edu

**Keywords:** α_2_-adrenergic receptors, yohimbine, clonidine, dexmedetomidine, emesis, least shrew, dorsal vagal complex

## Abstract

In contrast to cats and dogs, here we report that the α_2_-adrenergic receptor antagonist yohimbine is emetic and corresponding agonists clonidine and dexmedetomidine behave as antiemetics in the least shrew model of vomiting. Yohimbine (0, 0.5, 0.75, 1, 1.5, 2, and 3 mg/kg, i.p.) caused vomiting in shrews in a bell-shaped and dose-dependent manner, with a maximum frequency (0.85 ± 0.22) at 1 mg/kg, which was accompanied by a key central contribution as indicated by increased expression of c-*fos*, serotonin and substance P release in the shrew brainstem emetic nuclei. Our comparative study in shrews demonstrates that clonidine (0, 0.1, 1, 5, and 10 mg/kg, i.p.) and dexmedetomidine (0, 0.01, 0.05, and 0.1 mg/kg, i.p.) not only suppress yohimbine (1 mg/kg, i.p.)-evoked vomiting in a dose-dependent manner, but also display broad-spectrum antiemetic effects against diverse well-known emetogens, including 2-Methyl-5-HT, GR73632, McN-A-343, quinpirole, FPL64176, SR141716A, thapsigargin, rolipram, and ZD7288. The antiemetic inhibitory ID_50_ values of dexmedetomidine against the evoked emetogens are much lower than those of clonidine. At its antiemetic doses, clonidine decreased shrews’ locomotor activity parameters (distance moved and rearing), whereas dexmedetomidine did not do so. The results suggest that dexmedetomidine represents a better candidate for antiemetic potential with advantages over clonidine.

## 1. Introduction

Vomiting (emesis) is a protective reflex [1] which helps to remove ingested toxins from the gastrointestinal tract (GIT) [2]. The emetic loci include the nucleus tractus solitarius (NTS), the dorsal motor nucleus of the vagus (DMNX), and the area postrema (AP) in the brainstem, as well as the enteric nervous system (ENS) and enterochromaffin cells (EC cells) in the GIT [3]. The functional pathophysiology of vomiting indicates that emetic processes are controlled via interactions between the gastrointestinal enteric nervous system, the vagus, and the central nervous system (CNS) [3]. The emetic response involves multiple neurotransmitters/mediators such as dopamine (DA), serotonin (5-HT), substance P (SP), acetylcholine, histamine, prostaglandins, leukotrienes, and opiates, several of which have been recognized as mediators of chemotherapy-evoked vomiting [3,4,5]. The act of emesis may result from either direct activation of the brainstem dorsal vagal complex (DVC) emetic nuclei including the AP, NTS, and DMNX via a blood-borne pathway, and/or indirect stimulation of the DVC via activation of peripheral emetic loci such as the neurons of the ENS and release of emetic neurotransmitters from the gastrointestinal EC cells which subsequently activate the gastrointestinal vagal afferents to the brainstem [6,7].

α_2_-Adrenergic receptors are expressed both presynaptically (auto-receptors) and postsynaptically [8,9,10] in terminal fields in the nervous system, as well as on platelets which regulate several key physiological processes, including blood pressure, platelet aggregation, insulin secretion, lipolysis, and neurotransmitter release [11]. Moreover, activation or blockade of presynaptic α_2_-adrenergic receptors respectively suppress or enhance norepinephrine (NE) release. In contrast, stimulation or antagonism of postsynaptic α_2_-adrenergic receptors exert direct inhibitory or excitatory postsynaptic effects, respectively [12,13]. Systemic α_2_-adrenergic receptor perturbation can affect NE neurons in the nucleus of the solitary tract [14,15], as well as many α_2_-adrenergic receptor-expressing regions in the nervous system [16,17,18].

α_2_-Adrenergic receptor agonists (e.g., clonidine, xylazine) have been shown to evoke emesis in both cats and dogs [19,20,21,22,23,24,25,26]. Moreover, clonidine appears to be the most potent emetic α_2_-adrenergic receptor agonist with an ED_50_ of 25 µg/kg (i.m.) in dogs and 0.075 µg/kg (i.c.v.) in cats [20,21]. The evoked emesis is thought to be mediated by α_2_-adrenergic receptors, since it was blocked by the prototypical monoterpenoid indole alkaloid α_2_-adrenergic receptor antagonist, yohimbine [19,20,21]. In addition, several studies also demonstrate that dexmedetomidine can evoke dose-dependent emesis in cats and dogs [26,27,28,29]. However, in contrast to its emetic effects in cats and dogs, clonidine failed to trigger vomiting even at large doses in ferrets [30], pigeons [31], or least shrews (present study). On the contrary, clonidine was found to prevent vomiting induced by type 4 phosphodiesterase (PDE4) inhibitors in ferrets, and reserpine-evoked emesis in pigeons, whereas administration of yohimbine alone in both species produced unexpected vomiting [30,31]. Moreover, similar emetic responses were observed in ferrets following administration of either peripherally acting α_2_-adrenergic receptor antagonist MK-467, or its brain-penetrating analog MK-912, indicating both a peripheral and a central locus of action [30]. In humans, nausea and vomiting are also common side-effects of parenterally administered yohimbine [32]. In the clinical setting, a number of meta-analysis studies show that the incidence of post-operative nausea and vomiting (PONV) can be significantly reduced by both clonidine [33,34,35,36] and dexmedetomidine [36,37] in patients.

Shrews are assigned to the order of Insectivora and are among the earliest animals. Shrews are considered closer to primates than rodents, lagomorphs, and carnivores in the phylogenetic system [38]. Unlike the large emetic animal models discussed above, the least shrew (*Cryptotis parva*) is relatively much smaller (3–6 g in weight), and it is a well-validated experimental emetic model that has been used to test the emetic and antiemetic potential of diverse drugs [6,39].

Due to the discussed species differences, the present study sought to investigate the emetic/antiemetic potential of α_2_-adrenergic receptor ligands in the least shrew animal model of vomiting. Thus, we initially investigated whether intraperitoneal (i.p.) administration of varying doses of clonidine, dexmedetomidine, or yohimbine can evoke vomiting in least shrews. We found that the α_2_-adrenergic receptor agonists clonidine and dexmedetomidine do not induce vomiting in least shrews, whereas its corresponding antagonist yohimbine evoked emesis in a dose-dependent and bell-shaped manner. Secondly, we examined via immunohistochemistry whether the dorsal vagal emetic loci in the least shrew brainstem mediate the emetic effect of a maximally effective dose of yohimbine (1.0 mg/kg, i.p.) through induction of c-*fos*, 5-HT-, and/or SP-release. Thereafter, we further investigated whether clonidine and dexmedetomidine display broad-spectrum antiemetic efficacy against diverse emetogens, such as selective agonists of serotonin 5-HT_3_ (2-Methyl-5-HT)-, substance P (SP) neurokinin NK_1_ (GR73632)-, muscarinic M_1_ (McN-A-343), and dopamine D_2_ (quinpirole)-receptors, as well as the cannabinoid CB_1_ receptor inverse agonist/antagonist SR141716A, and Ca^2+^ channel modulators, including the selective L-type Ca^2+^ channel (LTCC) agonist FPL6417 and the sarco/endoplasmic reticulum Ca^2+^-ATPase (SERCA) inhibitor thapsigargin, which evoke pronounced vomiting in least shrews [40,41]. We also investigated the antiemetic potential of clonidine and dexmedetomidine against the PDE4 inhibitor rolipram- [30,42] and the hyperpolarization-activated cyclic nucleotide-gated (HCN) channel blocker ZD7288-induced vomiting [43].

## 2. Results

### 2.1. Dose-Response Emetic Effect of Yohimbine in Least Shrews

Intraperitoneal injection of yohimbine (3 mg/kg) can evoke vomiting in all tested ferrets [30]. In the present study, we initially assessed the emetic/antiemetic potential of clonidine (0, 0.1, 1, 5, and 10 mg/kg, i.p., *n* = 6–10 per group), dexmedetomidine (0, 0.01, 0.05, 0.1, 0.5, and 1 mg/kg, i.p., *n* = 6–10 per group), and yohimbine (0, 0.5, 0.75, 1, 1.5, 2, and 3 mg/kg, i.p., *n* = 7–14 per group) in the least shrew. Only yohimbine (0, 0.5, 0.75, 1, 1.5, 2, and 3 mg/kg, i.p.) increased both the frequency of emesis and the percentage of shrews vomiting in a bell-shaped and dose-dependent manner (Figure 1A,B). A Kruskal–Wallis (KW) non-parametric ANOVA test showed that, relative to the vehicle-pretreated control group, yohimbine significantly increased the mean vomiting frequency (KW (6, 65) = 14.16; *p* = 0.0279) with a maximum (0.85 ± 0.22) occurring at 1 mg/kg (*p* = 0.0259; Dunn’s test). At a dose of 3 mg/kg, few vomits were observed (Figure 1A). The chi-square test indicated that the percentage of animals exhibiting emesis in response to yohimbine also showed a bell-shaped and dose-dependent increase (*χ*^2^ (6, 65) = 9.276, *p* = 0.1586), but a significant difference was only observed at the 1 mg/kg dose, which evoked vomits in 61.54% of tested shrews (*p* = 0.0445; chi-square test; Figure 1B).

As shown in Figure 1C, the α_2_-adrenergic receptor agonist clonidine dose-dependently attenuated the frequency of yohimbine (1 mg/kg, i.p.)-induced vomiting (KW (4, 42) = 10.64; *p* = 0.0309) with an ID_50_ value of 0.205 (0.01437–1.882) mg/kg. Clonidine completely prevented the evoked vomiting at its 10 mg/kg dose (*p* = 0.0133). Clonidine also reduced the percentage of shrews vomiting (*χ*^2^ (4, 42) = 10.98, *p* = 0.0268) with an ID_50_ value of 0.2093 (0.02186–1.626) mg/kg, with significant reduction occurring at its 5 mg/kg (*p* = 0.0428) and complete protection at 10 mg/kg (*p* = 0.0048) (Figure 1D). Likewise, dexmedetomidine (0, 0.01, 0.05, and 0.1 mg/kg) reduced both the frequency (KW (3, 34) = 8.073; *p* = 0.0445) and percentage (*χ*^2^ (3, 34) = 8.78; *p* = 0.0324) of shrews vomiting in response to yohimbine (1 mg/kg, i.p.) in a dose-dependent manner with ID_50_ values of 0.02113 (0.002887–0.0981) mg/kg and 0.02253 (0.004506–0.08576) mg/kg, respectively (Figure 1E,F). Dexmedetomidine at 0.1 mg/kg completely suppressed both the mean frequency (*p* = 0.0177) and the percentage (*p* = 0.0048) of shrews vomiting in response to yohimbine (1 mg/kg, i.p.; Figure 1E,F).

### 2.2. Yohimbine Activates the Brainstem Emetic Nuclei

Since the 1 mg/kg (i.p.) dose of yohimbine caused maximal frequency of emesis in the tested shrews, we conducted immunohistochemistry to determine c-*fos* responsiveness following intraperitoneal administration of this dose of yohimbine. Figure 2A,B show that very few c-*fos* positive cells were observed in the DVC emetic nuclei in shrew brainstem sections from vehicle-treated controls, with mean values of 10.33 ± 1.65, 56.5 ± 4.99, and 9.67 ± 1.15 in the AP, NTS, and DMNX, respectively (Figure 2E). Relative to the vehicle-treated control group, a 1 mg/kg (i.p.) dose of yohimbine caused significant increases in c-*fos* expression in the brainstem throughout the three DVC emetic nuclei. Following vomiting induced by yohimbine, the average numbers of c-*fos* positive cells were increased to 50.17 ± 3.98 in the AP (*p* < 0.0001 vs. Vehicle), 166.3 ± 15.41 in the NTS (*p* < 0.0001 vs. Vehicle), and 36.17 ± 4.73 in DMNX (*p* = 0.0003 vs. Vehicle), respectively (Figure 2C–E).

### 2.3. Effect of Yohimbine on 5-HT- and SP-Release in Brainstem Emetic Nuclei

Our previous studies have shown that emetogens (e.g., cisplatin, FPL64176, and thapsigargin)-evoked emesis were accompanied by increases in 5-HT or SP immunoreactivity in brainstem emetic nuclei [41,44,45]. In the current study, we tested whether yohimbine administration (1 mg/kg., i.p.) could also increase 5-HT and SP immunoreactivity in brainstem DVC containing emetic nuclei AP, NTS, and DMNX. Least shrews were euthanized at 15 min and 30 min post yohimbine treatment and were subjected to immunohistochemistry to label 5-HT and SP. A representative image (Figure 3A) of the brainstem DVC emetic nuclei after 5-HT immunolabelling shows that in vehicle-treated control group, the highest density of 5-HT-positive fibers are in the dorsomedial subdivision of the NTS. A lower density of the 5-HT immunoreactive profile is noted in the adjacent subnuclei of NTS and DMNX. The AP had a few 5-HT-containing neurons. Figure 3D,J show that shrews exhibited significant increases in 5-HT immunoreactivity in the AP (*p* = 0.0337 vs. vehicle), NTS (*p* = 0.0298 vs. vehicle), and DMNX (*p* = 0.0446 vs. vehicle) at 15 min post yohimbine treatment. Figure 3G,J indicate that at 30 min post-injection, 5-HT immunoreactivity in the NTS, DMNX, and AP returned to basal levels.

Representative images acquired from SP immunolabeling are presented as Figure 3B,E,H. In the brainstem DVC of the vehicle-treated control group (Figure 3B), SP-immunoreactive fibers were found in highest concentration within the DMNX and to a lesser extent in the NTS, but rarely in the AP. Figure 3E,K show that yohimbine caused a significant increase in SP immunoreactivity only in the DMNX region of the shrew DVC (*p* = 0.0371 vs. vehicle) at 15 min post yohimbine treatment. Figure 3H,K show the intensity of SP staining in the DMNX returned to basal level at 30 min post yohimbine injection. Following 5-HT and SP labeling, sections were counterstained with DAPI to visualize cellular nuclei (Figure 3C,F,I).

### 2.4. The Broad-Spectrum Antiemetic Potential of the α_2_-Adrenergic Receptor Agonists Clonidine and Dexmedetomidine against Vomiting Evoked by Diverse Receptor-Selective Emetogens

Figure 4 and Figure 5 demonstrate the broad-spectrum antiemetic potential of both clonidine and dexmedetomidine against vomiting evoked by diverse receptor-selective emetogens, respectively. In fact, pretreatment with clonidine (0, 0.1, 1, 5, and 10 mg/kg, *n* = 7–10 per group), reduced both the frequency (KW (4, 40) = 26.40; *p* < 0.0001) and percentage (*χ*^2^ (4, 40) = 26.52; *p* < 0.0001) of shrews vomiting in a dose-dependent manner in response to the administration of the 5-HT_3_ receptor-selective agonist, 2-Methyl-5-HT (5 mg/kg, i.p.), with respective ID_50_ values of 0.6431 (0.224–1.494) mg/kg and 1.353 (0.7354–2.368) mg/kg (Figure 4A,B). The mean frequency of 2-Methyl-5-HT-induced emesis was significantly reduced at its 5 mg/kg dose (*p* = 0.0003) and with complete suppression at its 10 mg/kg dose (*p* = 0.0003). Significant decreases in the percentage of shrews vomiting were also noted at its 5 mg/kg dose (90%; *p* < 0.0001) and complete protection at 10 mg/kg dose (100%; *p* < 0.0001). Likewise, shrews pretreated with dexmedetomidine (0, 0.01, 0.05, and 0.1 mg/kg, *n* = 8–10 per group) also reduced both the frequency (KW (3, 31) = 8.331; *p* = 0.0397) and percentage (*χ*^2^ (3, 31) = 10.79; *p* = 0.0129) of shrews vomiting in a dose-dependent manner in response to 2-Methyl-5-HT (5 mg/kg, i.p.) with ID_50_ values of 0.02641 (0.007458–0.07622) mg/kg and 0.05137 (0.02349–0.1112) mg/kg, respectively (Figure 5A,B). Dexmedetomidine at 0.1 mg/kg significantly suppressed both the mean frequency (*p* = 0.0124) and the percentage (75%; *p* = 0.005) of shrews vomiting in response to 2-Methyl-5-HT (5 mg/kg, i.p.).

Next, the antiemetic effect of varying doses of clonidine and dexmedetomidine were tested against vomiting caused by the NK_1_ receptor selective agonist GR73632 (5 mg/kg, i.p.). As shown in Figure 4C, clonidine (0, 0.1, 1, 5, and 10 mg/kg, *n* = 7–10 per group) also caused dose-dependent decreases in the mean frequency (KW (4, 39) = 22.46; *p* = 0.0002) of vomits induced by GR73632 with significant reductions occurring at its 1 mg/kg (*p* = 0.0108), 5 mg/kg (*p* = 0.005), and 10 mg/kg (*p* = 0.0002) doses, and with an ID_50_ value of 0.333 (0.1035–0.9538) mg/kg. The percentage of shrews vomiting was also reduced in a dose-dependent fashion (*χ*^2^ (4, 39) = 12.51; *p* = 0.0139) with significant reductions at its 5 (33.33%; *p* = 0.0466) and 10 mg/kg (71.43%; *p* = 0.0015) doses, and with an ID_50_ value of 6.226 (2.959–13.5) mg/kg (Figure 4D). Figure 5C,D show that dexmedetomidine (0, 0.01, 0.05, and 0.1 mg/kg, *n* = 7–9 per group) dose-dependently suppressed vomiting caused by GR73632 (5 mg/kg, i.p.). The frequency of GR73632-induced emesis (KW (3, 28) = 8.195; *p* = 0.0421) was significantly reduced at 0.1 mg/kg (*p* = 0.0156) with an ID_50_ value of 0.04907 (0.0164–0.1427) mg/kg (Figure 5C). Likewise, a significant decrease (*χ*^2^ (3, 28) = 9.429; *p* = 0.0241) in the percentage of animals vomiting were also noted at its 0.1 mg/kg dose (62.5%; *p* = 0.0048) with an ID_50_ value of 0.08766 (0.04151–0.2017) mg/kg (Figure 5D).

Next, the antiemetic effect of clonidine and dexmedetomidine were assessed against vomiting caused by the muscarinic M_1_ receptor agonist, McN-A-343 (2 mg/kg, i.p.). Figure 4E,F show that clonidine (0, 0.1, 1, 5, and 10 mg/kg, *n* = 6–9 per group) dose-dependently suppressed vomiting (KW (4, 34) = 18.96; *p* = 0.0008) caused by McN-A-343 (2 mg/kg, i.p.). The frequency of McN-A-343-induced emesis was significantly reduced at its 5 mg/kg dose (*p* = 0.0084) and completely reduced at its 10 mg/kg dose (*p* = 0.001) with an ID_50_ value of 0.6007 (0.1618–1.76) mg/kg (Figure 4E). Likewise, significant decreases (χ^2^ (4, 34) = 20.92; *p* = 0.0003) in the percentage of animals vomiting were also noted at its 5 mg/kg dose (75%; *p* = 0.0019) with complete protection at the 10 mg/kg dose (100%; *p* = 0.0002), having an ID_50_ value of 1.588 (0.744–3.202) mg/kg (Figure 4F). Figure 5E,F show that dexmedetomidine (0, 0.01, 0.05, and 0.1 mg/kg, *n* = 7–8 per group) also dose-dependently suppressed vomiting caused by McN-A-343 (2 mg/kg, i.p.). The frequency of McN-A-343-induced emesis was significantly reduced (KW (3, 27) = 11.11; *p* = 0.0111) at 0.05 mg/kg (*p* = 0.0166) and 0.1 mg/kg (*p* = 0.0089) with an ID_50_ value of 0.02732 (0.009468–0.06793) mg/kg (Figure 5E). However, dexmedetomidine failed to significantly protect shrews from vomiting (χ^2^ (3, 27) = 6.285; *p* = 0.0985) having an ID_50_ value of 0.08931 (0.03922–0.2283) mg/kg (Figure 5F).

Thereafter, the antiemetic effect of varying doses of clonidine (0, 0.1, 1, 5, and 10 mg/kg, *n* = 10 per group) were examined against vomiting caused by the dopamine D_2/3_ receptor agonist quinpirole (2 mg/kg, i.p.; Figure 4G,H). The mean frequency of quinpirole-induced emesis (KW (4, 45) = 17.19; *p* = 0.0018) was significantly reduced by clonidine at its 1 mg/kg (*p* = 0.025), 5 mg/kg (*p* = 0.0051) and 10 mg/kg (*p* = 0.0013) doses, with an ID_50_ value of 0.7356 (0.1996–2.239) mg/kg (Figure 4G). However, there was not a significant decrease (χ^2^ (4, 45) = 9.201; *p* = 0.0563) in the percentage of animals vomiting with an ID_50_ value of 3.934 (1.304–10.17) mg/kg (Figure 4H). Pretreatment with dexmedetomidine (0, 0.01, 0.05, and 0.1 mg/kg, *n* = 9–11 per group) also dose-dependently suppressed the mean frequency of vomiting (KW (3, 36) = 11.31; *p* = 0.0102) caused by quinpirole (2 mg/kg, i.p.; Figure 5G). Significant decreases in the frequency of vomiting were noted at its 0.05 (*p* = 0.0188) and 0.1 mg/kg (*p* = 0.0079) doses with an ID_50_ value of 0.05819 (0.02685–0.1258) mg/kg (Figure 5G). The percentage of shrews vomiting (χ^2^ (3, 36) = 11.72; *p* = 0.0084) was significantly reduced at its 0.1 mg/kg dose (36.36%, *p* = 0.0341) with an ID_50_ value of 0.276 (0.1526–0.6753) mg/kg (Figure 5H).

We also investigated the antiemetic potential of clonidine and dexmedetomidine against vomiting evoked by the CB_1_ receptor inverse agonist/antagonist SR141716A (20 mg/kg, i.p.). As shown in Figure 4I, clonidine (0, 0.1, 1, 5, and 10 mg/kg, *n* = 8–10 per group) caused a dose-dependent decrease in the frequency of SR141716A-evoked vomits (KW (4, 41) = 23.22; *p* = 0.0001) with significant reductions occurring at its 1 mg/kg (*p* = 0.0244), 5 mg/kg (*p* = 0.0022), and 10 mg/kg (*p* < 0.0001) doses, and with an ID_50_ value of 0.0875 (0.02819–0.2546) mg/kg. The chi-square test indicates that the number of shrews vomiting in response to SR141716A was also significantly attenuated by clonidine (χ^2^ (4, 41) = 19.1; *p* = 0.0008) with 40% protection at its 0.1 mg/kg (*p* = 0.0253) and 1 mg/kg (*p* = 0.0253) doses, 62.5% protection at 5 mg/kg (*p* = 0.0033), and complete protection at 10 mg/kg (*p* < 0.0001) with an ID_50_ value of 1.291 (0.3871–3.31) mg/kg (Figure 4J). Figure 5I,J show that dexmedetomidine (0, 0.01, 0.05, and 0.1 mg/kg, *n* = 8–9 per group) also dose-dependently suppressed vomiting caused by SR141716A (20 mg/kg, i.p.). The mean frequency of SR141716A-induced emesis was significantly reduced (KW (3, 29) = 13.21; *p* = 0.0042) at 0.1 mg/kg (*p* = 0.004) with an ID_50_ value of 0.02453 (0.007966–0.06693) mg/kg (Figure 5I). The percentage of shrews vomiting (χ^2^ (3, 29) = 17.81; *p* = 0.0005) was significantly reduced at its 0.05 mg/kg (62.5%, *p* = 0.007)- and 0.1 mg/kg (75%, *p* = 0.0019) doses with an ID_50_ value of 0.04303 (0.02993–0.08087) mg/kg (Figure 5J).

### 2.5. The Antiemetic Potential of the α_2_-Adrenergic Receptor Agonists Clonidine and Dexmedetomidine against Vomiting Evoked by Ca^2+^ Channel Regulators

We then investigated the antiemetic effect of clonidine and dexmedetomidine against vomiting evoked by Ca^2+^ channel regulators, such as the LTCC agonist FPL64176 (10 mg/kg, i.p.) and the SERCA inhibitor thapsigargin (0.5 mg/kg, i.p.) (Figure 6 and Figure 7).

Pretreatment with clonidine (0, 0.1, 1, 5, and 10 mg/kg, *n* = 6–9 per group) significantly attenuated the mean frequency of FPL64176-induced vomiting in a dose-dependent manner (KW (4, 34) = 19.15; *p* = 0.0007) with significant reductions occurring at its 5 mg/kg (*p* = 0.01) and 10 mg/kg (*p* = 0.0008; ID_50_ value: 0.404 (0.1078–1.332) mg/kg; Figure 6A) doses. In addition, the percentage of shrews vomiting in response to FPL64176 was also suppressed by clonidine (χ^2^ (4, 34) = 18.34; *p* = 0.0011) at its 5 mg/kg dose (66.67%, *p* = 0.0042) and completely protected at its 10 mg/kg dose (100%, *p* = 0.0001) with ID_50_ value of 1.661 (0.6772–3.796) mg/kg (Figure 6B). Dexmedetomidine (0, 0.01, 0.05, and 0.1 mg/kg, *n* = 7–11 per group) also dose-dependently suppressed vomiting caused by FPL64176 (KW (3, 35) = 16.43; *p* = 0.0009) with significant reductions occurring at its 0.05 (*p* = 0.0028) and 0.1 mg/kg (*p* = 0.0009) doses, with an ID_50_ value of 0.006993 (0.001346–0.02088) mg/kg (Figure 7A). Likewise, significant decreases (*χ*^2^ (3, 35) = 11.46; *p* = 0.0095) in the percentage of animals vomiting were also noted at its 0.05 (54.55%; *p* = 0.0057) and 0.1 mg/kg (71.43%; *p* = 0.0015) doses with an ID_50_ value of 0.03667 (0.01645–0.0796) mg/kg (Figure 7B).

Figure 6C,D show that clonidine (0, 0.1, 1, 5, and 10 mg/kg, *n* = 7–9 per group) dose-dependently suppressed vomiting caused by thapsigargin (0.5 mg/kg, i.p.). The frequency of thapsigargin-induced emesis (KW (4, 36) = 25.05; *p* < 0.0001) was significantly reduced at 1 mg/kg (*p* = 0.0457), 5 mg/kg (*p* = 0.0013), and 10 mg/kg (*p* = 0.0003) doses, with an ID_50_ value of 0.1488 (0.04487–0.4809) mg/kg (Figure 6C). Significant decreases in the percentage of animals vomiting (*χ*^2^ (4, 36) = 16.38; *p* = 0.0026) occurred at its 5 mg/kg (55.56%; *p* = 0.0121) and 10 mg/kg (66.67%; *p* = 0.0041) doses with an ID_50_ value of 4.691 (2.37–9.161) mg/kg (Figure 6D). As shown in Figure 7C,D, dexmedetomidine (0, 0.01, 0.05, and 0.1 mg/kg, *n* = 8–9 per group) also caused a dose-dependent decrease in the mean frequency of vomits (KW (3, 30) = 8.296; *p* = 0.0403) induced by the thapsigargin (0.5 mg/kg, i.p.) with a significant reduction at its 0.1 mg/kg dose (*p* = 0.0135; ID_50_ value: 0.05317 (0.017–0.2617) mg/kg; Figure 7C), but no significant decrease (*χ*^2^ (3, 30) = 5.542; *p* = 0.1362) in the percentage of animals vomiting was observed (ID_50_ value: 0.1372 (0.06515–0.363) mg/kg; Figure 7D).

### 2.6. The Antiemetic Potential of Clonidine and Dexmedetomidine against Vomiting Evoked by Rolipram, the Inhibitor of PDE4 in Shrews

In the ferret, clonidine was found to prevent emesis induced by PDE4 inhibitors, such as PMNPQ (i.e., 6-(4-pyridylmethyl)-8-(3-nitrophenyl) quinoline, rolipram, and CT–2450 (i.e., (R)-N-{4-[1-(3-cyclopentyloxy-4-methoxyphenyl)-2-(4- pyridyl)ethyl]phenyl}N9-ethylurea) [30]. In the present study, we explored the antiemetic potential of the α_2_-adrenergic receptor agonists clonidine and dexmedetomidine against rolipram (1 mg/kg, i.p.)-induced vomiting in shrews [42].

As shown in Figure 8A, clonidine (0, 0.1, 1, 5, and 10 mg/kg, *n* = 8–10 per group) caused dose-dependent decreases in the frequency of rolipram-evoked vomits ((KW (4, 41) = 10.9; *p* = 0.0277) with significant reductions occurring at its 5 mg/kg (*p* = 0.0332) and 10 mg/kg (*p* = 0.0182) doses, with an ID_50_ value of 1.024 (0.1497–4.052) mg/kg. The percentage of animals vomiting in response to rolipram (χ^2^ (4, 41) = 11.21; *p* = 0.0243) was also significantly reduced at its 0.1 mg/kg (40%, *p* = 0.0253), 1 mg/kg (60%, *p* = 0.0034), 5 mg/kg (62.5%, *p* = 0.0033) and 10 mg/kg (62.5%, *p* = 0.0033) doses with an ID_50_ value of 0.7897 (0.1143–3.64) mg/kg (Figure 8B). Dexmedetomidine (0, 0.01, 0.05, and 0.1 mg/kg, *n* = 10 per group) also dose-dependently suppressed the mean vomiting frequency evoked by rolipram (KW (3, 36) = 10.7; *p* = 0.0135) with significant reductions at its 0.05 mg/kg (*p* = 0.0305) and 0.1 mg/kg (*p* = 0.0084) doses, and with an ID_50_ value of 0.01748 (0.004243–0.05612) mg/kg (Figure 8C). Significant decreases (*χ*^2^ (3, 36) = 9.167; *p* = 0.0272) in the percentage of animals vomiting were also noted at its 0.01 mg/kg (50%; *p* = 0.0098), 0.05 mg/kg (50%; *p* = 0.0098) and 0.1 mg/kg (60%; *p* = 0.0034) doses with an ID_50_ value of 0.03517 (0.011–0.09391) mg/kg (Figure 8D).

### 2.7. The Antiemetic Effect of Clonidine and Dexmedetomidine against Vomiting Evoked by the HCN Channel Blocker ZD7288

Our previous study has shown that the HCN channel blocker ZD7288 induces vomiting in a dose-dependent manner, with maximal efficacy of 100% at 1 mg/kg (i.p.) [43]. In the current study, the antiemetic potential of clonidine and dexmedetomidine on ZD7288 (1 mg/kg, i.p.)-induced vomiting was also examined (Figure 9). Pretreatment with clonidine (0, 0.1, 1, 5, and 10 mg/kg, *n* = 10 per group) significantly attenuated the mean frequency of ZD7288-induced vomiting in a dose-dependent manner (KW (4, 40) = 29.08; *p* < 0.0001) with significant reductions occurring at its 5 mg/kg (*p* = 0.0003) and 10 mg/kg (*p* < 0.0001; ID_50_ value: 0.4023 (0.1636–0.8948) mg/kg; Figure 9A) doses. Significant decreases in the percentage of animals vomiting (*χ*^2^ (4, 40) = 17.22; *p* = 0.0017) also occurred at its 5 mg/kg (44.44%; *p* = 0.0233) and 10 mg/kg (55.56%; *p* = 0.0085) doses with an ID_50_ value of 7.819 (4.363–14.69) mg/kg (Figure 9B). Administration of dexmedetomidine (0, 0.01, 0.05, and 0.1 mg/kg, *n* = 7–10 per group) also suppressed the mean frequency of vomiting evoked by ZD7288 (KW (3, 30) = 8.11; *p* = 0.0438) with a significant reduction at its 0.1 mg/kg dose (*p* = 0.0155), and with an ID_50_ value of 0.08931 (0.03732–0.2473) mg/kg (Figure 9C). However, dexmedetomidine (0, 0.01, 0.05, and 0.1 mg/kg) failed to significantly protect shrews from vomiting (χ^2^ (3, 30) = 3.974; *p* = 0.2643; Figure 9D).

### 2.8. Open-Field Locomotor Studies

Figure 10 shows the effects of corresponding vehicle, varying doses of clonidine (0.01, 0.1, 1, 5, and 10 mg/kg, i.p.), and dexmedetomidine (0.01, 0.05, 0.1, and 0.5 mg/kg, i.p.) on the locomotor activities of shrews using the open-field locomotor test. A one-way ANOVA analysis demonstrated that clonidine significantly decreased (*F*_5, 39_ = 5.255; *p* = 0.0009) the total distance moved by the shrews at its 1 mg/kg (*p* = 0.0015), 5 mg/kg (*p* = 0.0225), and 10 mg/kg (*p* = 0.0114) doses in the 30 min observation period (Figure 10A). Likewise, the frequencies of rearing behavior were also significantly attenuated (*F*_5, 39_ = 5.223; *p* = 0.0009) by clonidine at its 1 mg/kg (*p* = 0.0299), 5 mg/kg (*p* = 0.0013), and 10 mg/kg (*p* = 0.0014; Figure 10B) doses. Relative to the corresponding vehicle-pretreated control group, dexmedetomidine only significantly decreased the total distance moved (*F*_4, 32_ = 6.877; *p* = 0.0004) and the frequency of rearing behavior (*F*_4, 32_ = 4.758; *p* = 0.004) at its 0.5 mg/kg dose (*p* = 0.0002 for distance moved, Figure 10C; *p* = 0.0009 for rearing frequency, Figure 10D), which is larger than its antiemetic doses.

## 3. Discussion

### 3.1. Significance of the Present Study

Per the Introduction Section, currently there is significant confusion in the published literature on the ability of α_2_-adrenergic receptor ligands evoking emetic/antiemetic effects, which are probably due to: (i) species variation in the array of animal models (e.g., cats, dogs, ferrets, pigeons, least shrews, humans) employed for emesis studies in different laboratories, (ii) lack of published comprehensive studies investigating the emetic/antiemetic potential of α_2_-adrenergic receptor agonists and antagonists in a given laboratory using one emesis model, and (iii) full pharmacological assessment of α_2_-adrenergic receptor ligands against diverse emetogens. In the present study, we investigated the emetic/antiemetic potential of two α_2_-adrenergic receptor agonists (clonidine and dexmedetomidine), as well as an α_2_-adrenergic receptor antagonist yohimbine, in the least shrew animal model of vomiting. Only yohimbine increased both the mean frequency of emesis and the percentage of shrews vomiting. The evoked vomiting was bell-shaped and occurred in a dose-dependent manner with a maximal efficacy of 1 mg/kg. However, only 61.5% of least shrews vomited at the 1 mg/kg dose, whereas larger doses of yohimbine were less efficacious. Our immunohistochemical findings demonstrated that yohimbine (1 mg/kg, i.p.) causes significant increases in both c-*fos* expression and release of 5-HT and SP in the shrew brainstem DVC emetic nuclei. Furthermore, both α_2_-adrenergic receptor agonists not only suppressed yohimbine-induced vomiting in a dose-dependent manner, but also emesis evoked by: (1) selective agonists of diverse emetic receptors including serotonin 5-HT_3_ (2-Methyl-5-HT), SP neurokinin NK_1_ (GR73632), muscarinic M_1_ (MCN-A-343), and dopamine D_2/3_ (quinpirole); (2) the selective CB_1_ receptor inverse agonist/antagonist (SR141716A); (3) Ca^2+^ channel modulators including the LTCC agonist FPL64176 and the SERCA inhibitor thapsigargin; (4) the PDE4 inhibitor rolipram; and 5) the HCN channel blocker ZD7288. Clonidine at antiemetic doses (1, 5, and 10 mg/kg, i.p.) decreased the locomotor activity of shrews in the open-field test, whereas dexmedetomidine did not influence shrew locomotion at its tested antiemetic doses.

### 3.2. Emetic Effect of Yohimbine in the Least Shrew

Existing experimental evidence suggests a role for α_2_-adrenergic receptor in emesis. The α_2_-adrenergic receptor agonist clonidine has been shown to evoke vomiting in a dose-dependent fashion in both cats and dogs in an α_2_-adrenergic receptor antagonist (yohimbine)-sensitive manner [19,20,21]. However, clonidine was found to prevent vomiting caused by either rolipram (an inhibitor of PDE4) in ferrets [30], or reserpine (a monoamine releaser) in pigeons [31]. Moreover, yohimbine not only induced dose-dependent emesis in both ferrets and pigeons, but also potentiated reserpine-induced vomiting and reversed the inhibitory effect of clonidine against reserpine-evoked vomiting [30,31]. However, the discussed emetic studies in pigeons and ferrets were limited in scope since yohimbine’s full dose-response emetic effects were not evaluated, but 100% of both species vomited in response to an i.p. injection of its 0.5 or 3 mg/kg doses, respectively [30,31]. In the present study, yohimbine evoked vomiting in a dose-dependent, but bell-shaped manner, with a maximal efficacy of 0.85 ± 0.22 vomits at 1 mg/kg dose in 61.54% of tested shrews, whereas its larger doses were less efficacious. A similar bell-shaped dose-response effect has also been observed in aggression studies where smaller doses of yohimbine increased, and larger doses decreased aggressive behavior in rats [46]. The bell-shaped dose-response effects may be due to the non-selective nature of yohimbine since it does not exclusively bind to α_2_-adrenergic receptors as it also (i) has moderate to weak affinity for dopamine D_2_-, adrenergic α_1_-, and serotonergic 5-HT_1A_ receptors [47,48,49], and (ii) can release monoamines (NE, DA, and 5-HT) both in the periphery and the CNS [50,51].

### 3.3. Yohimbine-Induced Expression of c-fos and Release of 5-HT and SP in the DVC Central Emetic Loci

Induction of c-*fos* immunoreactivity is an indirect but classical tool to evaluate neuronal activation following peripheral agonist administration [52]. Our lab has also utilized c-*fos* induction in the least shrew brainstem DVC emetic nuclei to demonstrate central responsiveness to peripheral administration of diverse emetogens [41,53]. In the current study, relative to vehicle injection, vomiting induced by yohimbine (i.p.) was followed by increased expression of c-*fos* in all the DVC emetic nuclei (the AP, NTS, and DMNV) in the shrew brainstem. Likewise, we have previously shown that the nonspecific emetogen cisplatin and the more selective 5-HT_3_ receptor agonist 2-Methyl-5-HT can evoke vomiting in least shrews and c-*fos* expression in their corresponding DVC emetic nuclei AP, NTS, and DMNV [53]. Such a pattern of c-*fos* expression in the DVC emetic nuclei is not surprising since the AP, NTS, and DMNX are populated by neurons containing a number of emetic neurotransmitters (e.g., 5-HT, DA) and corresponding receptors, and systemic administration of yohimbine promotes their release both in the periphery and brainstem [50,51]. In fact, prior depletion of emetic monoamines with reserpine can completely prevent vomiting caused by yohimbine or reserpine in pigeons, suggesting both yohimbine and reserpine may induce emesis by releasing monoamines, albeit via different mechanisms [31].

5-HT is an important gastrointestinal signaling molecule. In the CNS it regulates appetite and emesis, and in the gastrointestinal tract 5-HT is an important mediator of sensation (e.g., nausea and emesis) between the intestine and the brainstem [54]. It can evoke vomiting via peripheral stimulation of 5-HT_3_ receptors in least shrews, whereas its brain penetrant analog 2-Methyl-5-HT involves both central and peripheral components [3,55]. We have previously demonstrated that the LTCC agonist FPL64176-induced emesis was accompanied by an increase in 5-HT immunoreactivity in the dorsomedial subdivision of the NTS [45]. In the present study, compared to vehicle pretreatment, a 15 min exposure to yohimbine resulted in a moderate enhancement of 5-HT immunoreactivity in the AP, NTS, and DMNX. Indeed, yohimbine has been shown to release emetic monoamines, including 5-HT, via blockade of presynaptic α_2_-adrenergic receptors [55,56,57,58,59]. In the current study, the yohimbine-evoked increases in 5-HT-positive fibers were most abundant in the least shrew dorsomedial area of the NTS, followed by smaller increases in the adjacent subnuclei of the NTS and DMNX, but the AP contained fewer 5-HT-containg neurons. In the current study, yohimbine also evoked an increase in SP immunoreactivity in the DMNX at 15 min post treatment, which is not surprising, since release of stimuli-evoked SP in the rat or rabbit spinal dorsal horn can be modulated by yohimbine [60,61]. The current finding agrees with our published study which demonstrated that a 15–30 min thapsigargin exposure (0.5 mg/kg, i.p.), can increase the SP tissue content up to ~3 times over the basal level in the least shrew DMNX [41]. The DMNX is known to send appropriate output to the gastrointestinal tract to alter gastric motility [53,62]. Thus, yohimbine-evoked enhancements in 5-HT and SP immunoreactivity in the DVC suggests that both 5-HT and SP are likely to be involved in the evoked emesis.

### 3.4. Effects of α_2_-Adrenergic Receptor Agonists Clonidine and Dexmedetomidine on Locomotor Activity

A major concern regarding α_2_-adrenergic receptor agonists is the possibility of sedation [63]. Although the affinity of dexmedetomidine is eight times greater than clonidine [64], similar low doses of both α_2_-adrenergic agonists (0.03–0.1 mg/kg, i.p.) reduce spontaneous locomotor activity in rodents [65,66,67,68]. Thus, we examined the motor suppressive effects of both clonidine and dexmedetomidine. Clonidine at 0.01 and 0.1 mg/kg doses did not significantly affect either spontaneous locomotor activity (i.e., distance moved) or the frequency of rearing behavior in least shrews, but significantly decreased both behaviors at its antiemetic doses (1, 5, and 10 mg/kg) in the 30 min observation period. This should not be surprising since, unlike rodents, least shrews are constantly on the move and rarely remain motionless. Some clinical findings suggest that although sedation is a well-known side-effect of clonidine [69], it does not appear to be related to its antiemetic property. For example, Kumar et al. [70] have shown that in elderly patients undergoing intraocular surgery, clonidine had no effect on emesis despite the fact that it induced more sedation. In contrast to clonidine, the tested antiemetic doses of dexmedetomidine (0.01, 0.05, and 0.1 mg/kg) in the current study did not significantly decrease either shrew motor parameters. Overall, our data provide evidence for the safe use of dexmedetomidine as an antiemetic drug at human doses equivalent to less than 0.1 mg/kg in least shrews.

### 3.5. Antiemetic Effects of α_2_-Adrenergic Receptor Agonists Clonidine and Dexmedetomidine against Yohimbine and Diverse Emetogens

In contrast to cats and dogs (see introduction), both the non-selective (clonidine) and the more selective α_2_-adrenergic receptor agonist dexmedetomidine failed to trigger vomiting in shrews even at doses 333–400 times greater than their corresponding ED_50_ values reported in dogs (clonidine: 25 µg/kg, i.m.; dexmedetomidine: 0.003 mg/kg, s.c.; [20,26]). Subsequently, we investigated the antiemetic potential of clonidine and dexmedetomidine, and found that both α_2_-adrenergic receptor agonists dose-dependently reduce to varying degrees both the frequency and percentage of shrews vomiting caused by diverse emetogens, including the following.
**The α_2_-adrenergic receptor antagonist yohimbine (1 mg/kg, i.p.):** Both clonidine and dexmedetomidine reduced the mean frequency and percentage of shrews vomiting in response to yohimbine in a dose-dependent manner with respective ID_50_ values ranging between 0.21 mg/kg and 0.021 mg/kg, respectively. Thus, dexmedetomidine appears to be ten times more potent antiemetic than clonidine against yohimbine-evoked emesis, and both agents completely protected shrews from vomiting at their maximal tested dose of 0.1 and 10 mg/kg, respectively. Likewise, dexmedetomidine is reported to be eight times more potent than clonidine in the clinic [71]. It is suggested that yohimbine-induced vomiting could be due to blockade of the inhibitory α_2_-adrenergic receptors, which is associated with the enhanced release of monoamines NE, DA, and 5-HT [50]. Clonidine overrides the ability of yohimbine to release monoamines, and thus prevents vomiting evoked by the monoamine releaser, reserpine [31]. These maximal tested doses of clonidine and dexmedetomidine in least shrews were subsequently used as their utmost tested doses against vomiting produced by the following emetogens.**The more selective and centrally/peripherally-acting 5-HT_3_ receptor agonist, 2-methyl-5-HT (5 mg/kg, i.p.):** 5-HT_3_ receptor selective antagonists, such as tropisetron [72] or palonosetron [40], can suppress vomiting caused by 2-Methyl-5-HT. Here, clonidine and dexmedetomidine suppressed the mean frequency of vomiting and protected shrews from emesis in a dose-dependent manner. In the latter case, dexmedetomidine was 25 times more potent than clonidine in suppressing 2-Methyl-5-HT-evoked vomiting. However, clonidine completely protected shrews from vomiting at its highest tested dose (10 mg/kg), whereas the largest dose of dexmedetomidine (0.1 mg/kg) only prevented 75% of shrews from vomiting. This difference probably reflects the nonselective nature of clonidine since the most potent and selective antagonist of 5-HT_3_ receptors, palonosetron (10 mg/kg, i.p.), could not completely prevent least shrews from 2-Methyl-5-HT-induced vomiting [40]. Ca^2+^ mobilization via extracellular Ca^2+^ influx through 5-HT_3_ receptors and L-type Ca^2+^ channels, and intracellular Ca^2+^ release via ryanodine receptors (RyRs) present on the endoplasmic reticulum (ER), initiate Ca^2+^-dependent sequential activation of CaMKIIa and signal-regulated kinase 1/2 (ERK_1/2_), which contribute to 2-Methyl-5-HT-evoked emesis [73]. The most studied signaling pathway of α_2_-adrenergic receptor is through G protein-coupled receptors, which consists of direct inhibition (“membrane-delimited”) by G protein βγ (G_βγ_) subunit complexes of Ca^2+^ entry through voltage-gated Ca^2+^ (Cav) channels [74]. For α_2_-adrenergic receptor, the L-type Cav channel is the main effector [75]. Typically, L-type calcium currents are maximally inhibited by 50% upon α_2_-adrenergic agonist application [74]. We suggest that the antiemetic effects of clonidine and dexmedetomidine on 2-Methyl-5-HT-evoked emesis is probably related to the inhibition of the L-type calcium currents.**The selective SP neurokinin NK_1_ receptor agonist GR73632 (5 mg/kg, i.p.):** At this dose, GR73632 causes robust vomiting in least shrews through the activation of SP neurokinin NK_1_ receptors [76], which can be completely blocked by the selective and potent NK_1_ receptor antagonist netupitant (10 mg/kg, i.p.) in least shrews [77]. In the present study, both clonidine and dexmedetomidine prevented GR73632-evoked emesis in a dose-dependent manner. However, the largest tested dose of clonidine (10 mg/kg, i.p.) reduced the frequency of GR73632-evoked vomiting by 94% and the percentage of least shrews vomiting by 71.4%. Also, dexmedetomidine at its maximum tested dose (0.1 mg/kg) significantly but partially attenuated both the mean vomit frequency (76.3%) and the percentage of least shrews vomiting (62.5%) in response to GR73632. Their ID_50_ values indicate that dexmedetomidine is more potent than clonidine in preventing GR73632-evoked vomiting. The NK_1_ receptor is G-protein coupled and can increase cytoplasmic Ca^2+^ concentration [78,79,80]. In fact, GR73632 evokes an increase in intracellular Ca^2+^ concentration via both Ca^2+^ release from intracellular Ca^2+^ stores, and extracellular Ca^2+^ influx through the transient receptor potential channels [79]. In addition, we have shown that LTCC blockers amlodipine and nifedipine also suppress vomiting caused by GR73632 in a dose-dependent manner [40]. Per our discussion above, we suggest that the antiemetic effects of clonidine and dexmedetomidine on GR73632-induced emesis may involve suppression of intracellular Ca^2+^ concentration.**The cholinergic M_1_ receptor agonist McN-A-343 (2 mg/kg, i.p.):** Clonidine reduced both the mean frequency and the percentage of least shrews vomiting in response to McN-A-343 in a dose-dependent fashion, with complete emesis protection occurring at its 10 mg/kg dose. However, dexmedetomidine significantly but partially reduced the mean frequency of vomiting (66.7%) at its tested maximal dose of 0.1 mg/kg but failed to significantly protect shrews from vomiting. McN-A-343 can activate PKC and PKA phosphorylates to enhance Ca^2+^ influx through LTCC channels [81,82]. Moreover, the L-type calcium antagonist nifedipine can significantly and completely prevent in a potent and dose-dependent manner both the percentage of shrews vomiting and the mean frequency of emesis evoked by McN-A-343 (2 mg/kg, i.p.) in the least shrew [83]. The antiemetic effects of clonidine and dexmedetomidine on McN-A-343-induced emesis may also involve suppression of intracellular Ca^2+^ mobilization.**The dopamine D_2/3_ receptor preferring agonist quinpirole (2 mg/kg, i.p.):** Clonidine only significantly but partially reduced the mean frequency of quinpirole-evoked vomiting by 80.4% at its 10 mg/kg maximal tested dose but failed to completely protect shrews from vomiting. Likewise, dexmedetomidine significantly reduced both the mean frequency (57.8% reduction at 0.1 mg/kg) and the percentage of least shrews vomiting (36.4% protection at 0.1 mg/kg) in response to quinpirole, but these reductions were only partial. Although the dopamine D_2_ receptor-preferring antagonist sulpiride can completely prevent apomorphine (2 mg/kg, i.p.)-induced vomiting in least shrews at 2 mg/kg (s.c.), but it cannot fully protect shrews from quinpirole (2 mg/kg, i.p.)-evoked emesis even up to 8 mg/kg [84]. Thus, it appears that quinpirole-evoked emesis cannot be easily subdued by sulpiride. Mechanistically, we have previously demonstrated the involvement of the PI3K/mTOR/Akt signaling pathway in dopamine D_2_ receptor-evoked vomiting [85]. Since the α_2_-adrenergic receptor agonist tizanidine can reduce the expression levels of PI3K/Akt [86], it is possible that clonidine and dexmedetomidine may also affect this signaling system to suppress the evoked vomiting, but this remains to be investigated.**The cannabinoid CB_1_ receptor-selective inverse agonist/antagonist SR141716A (20 mg/kg, i.p.):** SR141716A can induce vomiting in the least shrew at large doses (20–40 mg/kg, i.p.) which can be fully prevented by CB_1_ receptor agonists, including Δ^9^-THC [87]. Clonidine significantly reduced both the mean frequency and the percentage of shrews vomiting in response to SR141716A in a dose-dependent manner with complete emesis protection at it 10 mg/kg dose. Dexmedetomidine reduced the mean frequency of vomiting by 88% and protected shrews from vomiting by 75% at its maximal tested dose of 0.1 mg/kg. SR141716A increases the release and turnover of monoamines DA, NE, and 5-HT [88] in several brain regions of rodents and least shrews, and can enhance capsaicin-evoked release of SP in the mouse spinal cord [89]. As discussed earlier, since clonidine can prevent vomiting caused by the monoamine releaser reserpine [31], it probably prevents SR141716A-evoked vomiting via a similar mechanism.**The selective LTCC agonist FPL64176 (10 m/kg, i.p.):** The LTCC regulates extracellular Ca^2+^ influx into the cytosol [90]. FPL64176 is an extracellular Ca^2+^-mobilizing agent and evokes vomiting in all tested shrews at 10 mg/kg [40,83]. Clonidine significantly and completely reduced the mean vomiting frequency and the percentage of shrews vomiting in response to FPL64176 challenge in a dose-dependent manner. Dexmedetomidine significantly and dose-dependently reduced the mean vomit frequency by 92.3% following its maximal tested dose (0.1 mg/kg), and protected 71.4% of shrews from FPL64176 (10 mg/kg, i.p.)-induced vomiting. However, their ID_50_ values indicate that dexmedetomidine is more potent than clonidine in preventing FPL64176-evoked vomiting. As was discussed earlier, the antiemetic effects of clonidine and dexmedetomidine on FPL64176-evoked emesis may also involve inhibition of L-type calcium currents.**Specific inhibitor of the SERCA pump thapsigargin (0.5 mg/kg, i.p.):** The SERCA pump is a main mechanism that transports free cytosolic Ca^2+^ into endoplasmic reticulum (ER) Ca^2+^ stores. Release of Ca^2+^ from the ER stores into the cytosol occurs through the inositol trisphosphate (IP_3_)- and ryanodine (RyR)-receptor ion channels localized on the ER membrane [91,92]. Thapsigargin is a specific and potent inhibitor of SERCA pumps and causes a rapid elevation in cytosolic Ca^2+^ concentrations. Thapsigargin can also release Ca^2+^ from the ER stores into the cytosol via IP_3_ and RyR calcium channels [93,94,95]. Thapsigargin causes vomiting by triggering an initial elevation in the cytoplasmic Ca^2+^ concentration by inhibiting the SERCA as well as releasing Ca^2+^ from the ER into the cytoplasm via both RyR- and IP_3_-receptors (IP3Rs), which is followed by an extracellular Ca^2+^ influx through LTCCs prior to the intracellular activation of the Ca^2+^-CaMKII-ERK1/2 cascade [41]. In the present study, clonidine significantly reduced the mean frequency of thapsigargin-evoked vomiting by 93.3% at its 10 mg/kg dose, but only partially protected 66.7% of shrews at this maximal dose. Moreover, dexmedetomidine only reduced the mean frequency of the evoked vomiting by 70.8% at its 0.1 mg/kg maximal dose but failed to significantly protect shrews from vomiting. Since the LTCC inhibitor nifedipine can dose-dependently inhibit thapsigargin-induced vomiting [41], we speculate that clonidine and dexmedetomidine may partially suppress extracellular Ca^2+^ influx through LTCCs, which could then attenuate the above-discussed signaling cascade, leading to a limited reduction in thapsigargin-evoked vomiting.**The PDE4 inhibitor rolipram (1 mg/kg, i.p.):** PDE4 inhibitors prevent metabolism of second messengers such as cAMP and increase their tissue levels. They have procognitive and antidepressant properties [96]. The emetic effect of some PDE4 inhibitors is thought to be a consequence of inhibition of PDE4 and the subsequent increase in cAMP levels in the brainstem DVC [30,42]. PDE4 inhibitors may mimic the pharmacological effect of α_2_-adrenergic receptor antagonists, which elevate intracellular levels of cAMP in noradrenergic neurons [30]. In contrast, α_2_-adrenergic receptor activation decreases intracellular levels of cAMP in noradrenergic neurons. PDE4 inhibitors are thought to modulate the release of emetic mediators including 5-HT, SP, and noradrenaline which are involved in the onset of the emetic reflex [30]. We have previously shown that the PDE4 inhibitor rolipram (1 mg/kg, i.p.) evokes both vomiting as well as significant increases in shrew brainstem cAMP levels, while pretreatment with SQ22536, an inhibitor of adenylyl cyclase, prevented the evoked emesis [42]. In the present study, clonidine dose-dependently and significantly, albeit partially, reduced both the mean vomit frequency (80.3%) and the percentage (62.5%) of shrews vomiting in response to rolipram at its highest tested dose, 10 mg/kg. Likewise, dexmedetomidine produced similar dose-dependent but partial reductions in both the mean vomit frequency (73.7%) and the percentage of shrews vomiting (60%) at its maximal tested dose, 0.1 mg/kg. The antiemetic effect of clonidine and dexmedetomidine against rolipram-induced vomiting might be due to suppression of increased intracellular tissue levels of cAMP evoked by rolipram in the shrew brainstem. In fact, administration of cAMP analogs such as 8-chloro-cAMP cause vomiting in cancer patients [97].**The HCN blocker ZD7288 (1 mg/kg, i.p.):** The HCN channels are a class of voltage-gated ion-channels permeable to Na^+^ and K^+^ and constitutively open at voltages near the resting membrane potential [98,99]. The hyperpolarization-activated currents (I*_h_*) mediated by HCN channels elicit membrane depolarization toward a threshold for action potential generation, which plays a pivotal role in controlling neuronal excitability [98,100]. The HCN channel blocker ZD7288 can reduce apomorphine-induced conditioned taste aversion to saccharin preference and depress the excitability of the AP since it blocks HCN channel activation [101]. We have recently demonstrated that ZD7288 (1 mg/kg, i.p.) evokes both vomiting in a dose-dependent manner as well as a robust expression of c-*fos* and ERK_1/2_ phosphorylation in the shrew brainstem DVC, indicating a central contribution to the evoked vomiting [43]. In the present study, clonidine significantly and dose-dependently reduced ZD7288-evoked mean vomit frequency (92.4% reduction at 10 mg/kg), but only partially protected shrews from vomiting (55.6% protection at 10 mg/kg). Dexmedetomidine only significantly reduced the mean frequency (63.1% protection at 0.1 mg/kg) and failed to significantly protect shrews from ZD7288-evoked vomiting. Given the well-known negative coupling of α_2_-adrenergic receptors to adenylate cyclase via a heterotrimeric G protein [102], any reduction in the cAMP level would decrease I*_h_*. Dexmedetomidine via α_2_-adrenergic receptors activates G-protein-coupled K^+^ channels and inhibits I*_h_*, which leads to membrane hyperpolarization [103]. In addition, clonidine can directly inhibit I*_f_* current [11]. Hence, we speculate that administration of clonidine and dexmedetomidine reverse the inhibitory effect of ZD7288 on the HCN channels and consequently suppresses ZD7288-induced vomiting in shrews.

As discussed in the introduction section, although both clonidine and dexmedetomidine are used as antiemetics for suppression of PONV, to date no basic or clinical study has compared their antiemetic potential against diverse emetogens. In the present study their antiemetic ID_50_ values are comparatively determined against various emetogens, and dexmedetomidine appears to be 3–69 times more potent than clonidine, demonstrating that dexmedetomidine has more efficacious antiemetic potential at non-sedating doses.

## 4. Materials and Methods

### 4.1. Animals

A colony of adult least shrews between 45–60 days old and weighing between 4 and 6 g from the Western University of Health Sciences Animal Facilities were employed in this study. Shrews were housed in groups of 5–10 on a 14:10 light: dark cycle and were fed and watered ad libitum. Animal experiments were conducted in accordance with the principles and procedures of the National Institutes of Health Guide for the Care and Use of Laboratory Animals. All protocols were approved by the Institutional Animal Care and Use Committee of Western University of Health Sciences (Protocol number R20IACUC018). All efforts were made to minimize animal suffering and to reduce the number of animals used in the experiments.

### 4.2. Chemicals

The following drugs were used in the present study: clonidine, dexmedetomidine, yohimbine, GR73632, SR141716A, FPL64176, thapsigargin, and ZD7288 were purchased from Tocris (Minneapolis, MN, USA); McN-A-343, quinpirole HCl, and rolipram were purchased from Sigma-Aldrich (St. Louis, MO, USA); 2-methyl-serotonin maleate salt (2-Methyl-5-HT) was purchased from Santa Cruz Biotechnology (Dallas, TX, USA). SR141716A was dissolved to twice the stated drug dose in a 1:1:18 solution of ethanol:Emulphor™:0.9% saline and was then diluted further with an equal volume of saline. FPL64176 was dissolved in 25% DMSO in water. Thapsigargin was dissolved in 10% DMSO in distilled water. Other drugs were dissolved in distilled water. All drugs were administered at a volume of 0.1 mL/10 g of body weight. The doses and routes used for the emetogens were based upon previous publications from our laboratory [6,43,87].

### 4.3. Behavioral Emesis Studies

On the day of the experiment, both male and female shrews [43] were brought from the animal facility, randomly separated into individual cages, and allowed to adapt to the experimental condition for at least 2 h. Daily food was withheld 2 h prior to the start of the experiment, but shrews were given 4 mealworms each 30 min prior to emetogen injection to help identify wet vomits as described previously [39]. Experiments were performed between 8:00 a.m. and 5:00 p.m.

To identify the emetic/antiemetic potential of either the α_2_-adrenergic receptor agonists clonidine and dexmedetomidine, or the corresponding antagonist yohimbine, different groups of shrews were injected with varying doses of either the corresponding vehicle, clonidine (0.1, 1, 5, and 10 mg/kg, i.p., *n* = 6–10), dexmedetomidine (0.01, 0.05, 0.1, 0.5, and 1 mg/kg, i.p., *n* = 6–10), or yohimbine (0.5, 0.75, 1, 1.5, 2, and 3 mg/kg, i.p., *n* = 7–14). Immediately following the injection, each shrew was placed in the observation cage and the frequency of emesis was recorded for the next 30 min. Based on the results obtained, the corresponding vehicles, clonidine, or dexmedetomidine at all tested doses failed to evoke vomiting in least shrews. However, the i.p. administration of yohimbine produced vomiting in shrews in a dose-dependent and bell-shaped response manner, and a 1 mg/kg dose of yohimbine caused maximal mean frequency of emesis in shrews, and thus was chosen for subsequent immunofluorescence staining and drug interaction studies.

Since the α_2_-adrenergic receptor antagonist yohimbine caused vomiting, we initially tested the antiemetic potential of its corresponding agonists clonidine and dexmedetomidine in the following manner: different groups of shrews were pretreated with an injection of either corresponding vehicles or varying doses of clonidine (0.1, 1, 5, and 10 mg/kg, i.p.) or dexmedetomidine (0.01, 0.05, and 0.1 mg/kg, i.p.) at 0 min. Following 30 min, the pretreated shrews were challenged for vomiting with a maximally effective dose of yohimbine (1 mg/kg, i.p.). Each shrew was then placed in the observation cage and the frequency of vomiting was recorded for the next 30 min. Using the same procedure, we subsequently investigated the antiemetic potential of these α_2_-adrenergic receptor agonists against fully efficacious emetic dose of the following emetogens in separate experiments [40,41,43]: (1) the selective serotonin 5-HT_3_ receptor agonist 2-Methyl-5-HT (5 mg/kg, i.p.) [73]; (2) the selective SP neurokinin NK_1_ receptor agonist GR73632 (5 mg/kg, i.p.) [76,77]; (3) the muscarinic M_1_ receptor agonist McN-A-343 (2 mg/kg, i.p.) [83]; (4) the dopamine D_2/3_ preferring receptor agonist quinpirole (2 mg/kg, i.p.) [84]; (5) the CB_1_ receptor inverse agonist/antagonist SR141716A (20 mg/kg, i.p.) [87]; (6) the LTCC agonist FPL64176 (10 mg/kg, i.p.) [45]; (7) the SERCA inhibitor thapsigargin (0.5 mg/kg, i.p.) [41]; (8) the PDE4 inhibitor rolipram (1 mg/kg, i.p.) [42]; (9) the HCN channel blocker ZD7288 (1 mg/kg, i.p.) [43]. In the emesis studies the observer was unaware of treatment conditions. Each shrew was used once, then euthanized with isoflurane (3%) in the anesthesia chamber following the termination of each behavioral experiment.

### 4.4. Immunohistochemistry and Image Analysis

#### 4.4.1. c-*fos* Staining and Image Analysis

Immunohistochemistry of the least shrew brainstem (20 µm) was conducted as previously reported [43,77]. Following vehicle or yohimbine (1 mg/kg, i.p.) injection, vomiting shrews were subjected to c-*fos* staining (*n* = 6 shrews per group). Following 90 min after the first emesis occurred, shrews were deeply anesthetized with isoflurane (3%), then transcardially perfused with 0.01 M phosphate buffered saline (PBS) followed by ice cold 4% paraformaldehyde (pH 7.4) in 0.01 M PBS for 10 min. Brainstems were post-fixed in the same fixative for 2 h, then placed in 0.1 M PB containing 30% sucrose at 4℃ until they sank. The brainstem was cut in 20 μm sections using a cryostat (Leica, Bannockburn, IL, USA), and pre-incubated in the blocking buffer (0.01 M PBS containing 10% normal donkey serum and 0.3% Triton X-100) for 1 h at room temperature. The slices were then incubated in a rabbit anti-c-*fos* primary antibody (1:1000, ab190289, Abcam, Cambridge, UK) in 0.01M PBS containing 5% normal donkey serum, 0.05% sodium azide, and 0.3% Triton X-100 at 4℃ overnight. The sections were washed 3 times (10 min each) in PBS and incubated in an Alexa Fluor 594 donkey anti-rabbit secondary antibody (1:500, A-21207, Invitrogen, Waltham, USA) in 0.01 M PBS containing 0.3% Triton X-100 for 2 h at room temperature, then washed 3 times (10 min each) and were mounted and coverslipped with an anti-fade mounting medium containing DAPI (Vector Laboratories, Newark, USA). Tile-scanning images of the brainstem sections containing the brainstem emetic nuclei (AP/NTS/DMNV) were taken by a confocal microscope (Zeiss LMS 880, Oberkochen, Germany) at 1024 × 1024 pixels with Zen software using Plan-Apochromat 20×/0.8 M27 objective. Cytoarchitectonic differences in the AP, NTS, and DMNV of the least shrew brainstem have been described in our published studies [41,45,76]. For each animal, c-*fos* positive cells in the AP and both sides of NTS and DMNX from 3 sections at 90 μm intervals were counted manually by an experimenter blind to the experimental conditions. The average value was used in statistical analysis.

#### 4.4.2. 5-HT and SP Immunohistochemistry

Shrews (*n* = 5–6 shrews per group) were treated with either vehicle or yohimbine (1 mg/kg, i.p.) and rapidly anesthetized with isoflurane and subjected to perfusion at 15 min and 30 min post-treatment to examine 5-HT and SP immunoreactivity. The experimental procedure prior to staining was performed as described above for Section 4.4.1. c-*fos* Staining and Image Analysis. Coronal brainstem sections (20 μm) were blocked with 0.1 M PBS containing 10% donkey serum and 0.3% Triton X-100, then incubated overnight at 4 °C with a mix of goat anti-5-HT primary antibody (1:1000, ab66047, Abcam) and rat anti-SP primary antibody (1:400, MAB356, EMD Millipore, Burlington, VT, USA) in 0.1 M PBS containing 5% donkey serum and 0.3% Triton X-100. Sections were washed 3 times (10 min each) in PBS and incubated in a mix of Alexa Fluor 488 donkey anti-goat (1:500, ab150133, Abcam) and cy3-conjugated donkey anti-rat (1:500, AP189C, EMD Millipore) secondary antibody in 0.1 M PBS containing 0.3% Triton X-100 for 2 h at room temperature. After washing with PBS 3 times (10 min each), sections were mounted with anti-fade mounting medium containing DAPI (Vector Laboratories). Images for the DVC were acquired using a confocal microscope (Zeiss LMS 880) as described above. Fluorescence intensity (mean gray value) of 5-HT and SP values were acquired using ImageJ software, as described previously [45].

### 4.5. Locomotor Activity Studies

Because α_2_-adrenergic receptor agonists have potent analgesic, sedative/hypnotic properties in humans and in experimental animals [104,105,106,107], the locomotor activity parameters (i.e., total distance moved and rearing behavior) were measured in shrews following injection of varying doses of clonidine (0, 0.1, 1, 5, and 10 mg/kg, i.p.) or dexmedetomidine (0, 0.01, 0.05, 0.1, and 0.5 mg/kg, i.p.). The Ethovision (version XT 9) locomotion analysis and behavior recognition system (Noldus Information Technology, Wageningen, The Netherlands) was used, as reported previously [108]. The parameters of Ethovision were set to record two locomotor activities: (1) total distance moved in centimeters (spontaneous locomotor activity), and (2) rearing frequency, which was recorded when the shrew was standing upright with a 5% reduction in body surface area as seen by the overhead video camera.

On the day of the experiment, both male and female shrews were brought in their home cages to the experimental room and were allowed to acclimate to a semi-dark environment for 1 h. Each shrew was further acclimated in an empty white plastic observation cage (27.5 × 27.5 × 28 cm) for 1 h before testing. Different groups of shrews were injected with either corresponding vehicle or varying doses of clonidine (0.1, 1, 5, and 10 mg/kg, i.p., *n* = 6–10) or dexmedetomidine (0.01, 0.05, 0.1, and 0.5 mg/kg, i.p., *n* = 6–10) at 0 min. Following 30 min, each shrew was individually placed in a white observation cage of the same dimension, and the two motor parameters were recorded for 30 min by an overhead camera and data were analyzed using the Noldus software. The chamber was thoroughly cleaned with 70% ethanol and dried to eliminate animal odors between test sessions. Each shrew was used once and euthanized using isoflurane at the end of the experiment.

### 4.6. Statistical Analysis

Statistical analyses were done using Graphpad Prism 8 (Graphpad software Inc., San Diego, CA, USA). The frequencies of vomits were analyzed using the Kruskal–Wallis non-parametric one-way analysis of variance (ANOVA) followed by Dunn’s post hoc test and expressed as the mean ± SEM. The percentage of animals vomiting across treatment groups at different doses was compared using the chi-square test. The locomotor activities and the differences of 5-HT/SP mean gray values among groups were compared with an ordinary ANOVA test followed by Dunnett’s post hoc test. The numbers of c-*fos*-positive cells between two groups were tested by unpaired *t*-test. *p* < 0.05 was considered statistically significant. Sample size calculations were determined as described by Chow [109].

## 5. Conclusions

Our results demonstrate that the α_2_-adrenergic receptor antagonist yohimbine is pro-emetic in least shrews, and its corresponding agonists clonidine and dexmedetomidine possess broad-spectrum antiemetic potential. In fact, yohimbine caused vomiting in a bell-shaped and dose-dependent manner, accompanied by increased c-*fos* expression and release of 5-HT and SP in the brainstem DVC emetic nuclei. Clonidine and dexmedetomidine not only suppressed yohimbine-evoked emesis in a dose-dependent manner, but also vomiting caused by other well-investigated emetogens of varied mechanism of actions. The broad-spectrum antiemetic effects of dexmedetomidine occur at much lower doses than those of clonidine. Furthermore, the antiemetic doses (1, 5, and 10 mg/kg, i.p.) of clonidine decreased the least shrew locomotor activity parameters, whereas dexmedetomidine lacked motor suppressive behaviors. The current results suggest both clonidine and dexmedetomidine possess broad-spectrum antiemetic potential, but dexmedetomidine’s antiemetic ability occurs at non-sedative doses.

## Figures and Tables

**Figure 1 ijms-25-04603-f001:**
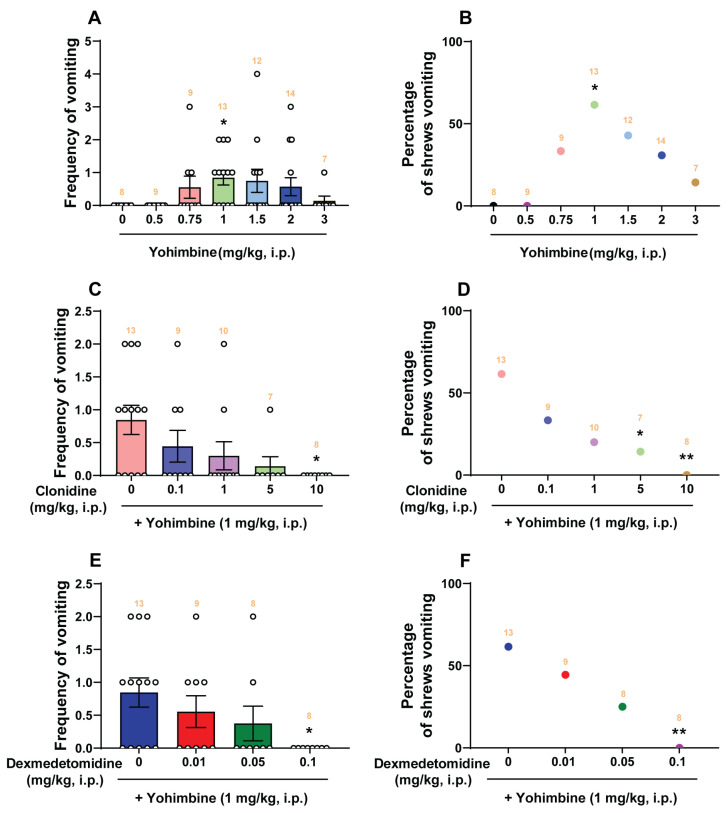
The pro-emetic effect of the α_2_-adrenergic receptor antagonist yohimbine and the corresponding antiemetic efficacy of the α_2_-adrenergic receptor agonists clonidine and dexmedetomidine in least shrews. Different groups of least shrews were given varying doses of yohimbine (0, 0.5, 0.75, 1, 1.5, 2, 3 mg/kg, i.p., *n* = 7–14 shrews per group (**A**,**B**). Emetic parameters were recorded for the next 30 min. In drug interaction studies, different groups of least shrews were given an injection (i.p.) of either the corresponding vehicle, or varying doses of clonidine (0.1, 1, 5, and10 mg/kg, i.p., *n* = 7–13 shrews per group (**C**,**D**) or dexmedetomidine (0.01, 0.05, and 0.1 mg/kg, i.p., *n* = 8–13 shrews per group (**E**,**F**), 30 min prior to an injection of yohimbine (1 mg/kg, i.p.), and were observed for the next 30 min. The frequency of emesis was analyzed with Kruskal–Wallis non-parametric one-way ANOVA followed by Dunn’s post hoc test and presented as mean ± SEM (**A**,**C**,**E**). The percentage of shrews vomiting was analyzed with chi-square test and presented as mean (**B**,**D**,**F**). * *p* < 0.05, ** *p* < 0.01 vs. 0 mg/kg. The number of animals in each group is presented on the top of the corresponding column.

**Figure 2 ijms-25-04603-f002:**
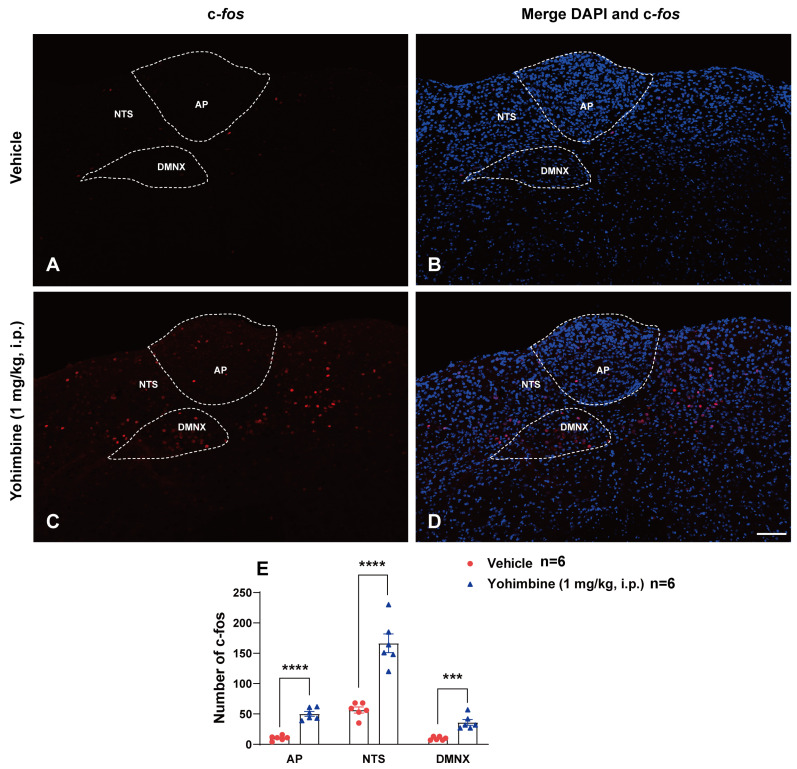
Immunohistochemical analysis of c-*fos* following emesis induced by systemic administration of the α_2_-adrenergic receptor antagonist yohimbine. Least shrews were sacrificed 90 min post vehicle treatment, or after the first vomiting occurred post systemic administration of yohimbine (1 mg/kg, i.p., *n* = 6 shrews per group). Shrew brainstem sections (20 μm) were stained with rabbit c-*fos* primary antibody and Alexa Fluor 594 donkey anti-rabbit secondary antibody. Nuclei were stained with DAPI in blue. Representative tile-scanned images (20×) show robust c-*fos* induction in the area postrema (AP), the nucleus tractus solitarius (NTS), and the dorsal motor nucleus of the vagus (DMNX) in response to yohimbine (1 mg/kg, i.p.; (**C**,**D**)) when compared to the vehicle-treated group (**A**,**B**). Scale bar = 100 μm. Quantified data show the yohimbine-induced c-*fos* expression in the AP, NTS, and DMNX in least shrew brainstem (**E**). Values represent the mean number of c-*fos*-positive nuclei of each region (AP/NTS/DMNV) per section and are presented as mean ± SEM (*n* = 6 shrews per group). *** *p* < 0.001, **** *p* < 0.0001 vs. Vehicle, Unpaired *t*-test.

**Figure 3 ijms-25-04603-f003:**
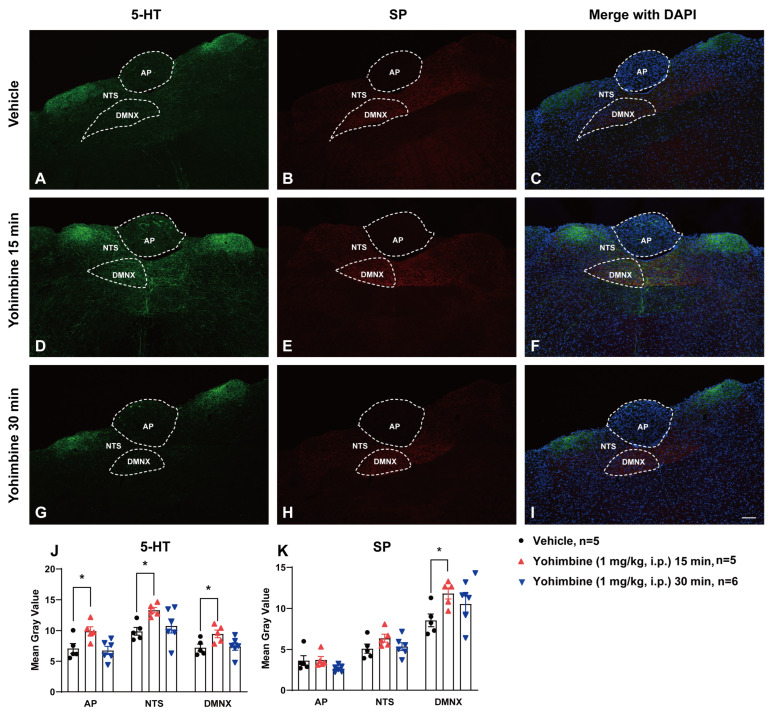
Immunohistochemical analysis of 5-HT and SP involvement following emesis induced by systemic administration of the α_2_-adrenergic receptor antagonist yohimbine. Shrews (*n* = 5–6 per group) were euthanized at 15 and 30 min post vehicle or yohimbine (1 mg/kg, i.p.) injection. Coronal brain sections (20 μm) containing the brainstem DVC emetic nuclei [area postrema (AP), nucleus of the solitary tract (NTS), and dorsal motor nucleus of the vagus (DMNX)] were immunolabeled with goat anti-5-HT antibody and rat anti-SP antibody overnight followed by Alexa Fluor 488 donkey anti-goat and cy3-conjugated donkey anti-rat secondary antibody incubation. After counterstaining with DAPI, images were acquired using a confocal microscope. Representative tile-scanned images (20×) images (**A**–**I**) show sections from vehicle, 15, and 30 min post yohimbine treated groups labeled with anti-5-HT (green), anti-SP antibodies (red), and merged with DAPI (blue), respectively. Scale bar = 100 μm. Quantified data show the yohimbine-induced release of 5-HT and SP in the AP, NTS, and DMNX regions of least shrew brainstem, respectively (**J**,**K**). Values represent the mean gray value of released 5-HT and SP of each region (AP/NTS/DMNV) per section and are presented as mean ± SEM (*n* = 5–6 shrews per group). * *p* < 0.05 vs. Vehicle, one-way ANOVA followed by Dunnett’s post hoc test.

**Figure 4 ijms-25-04603-f004:**
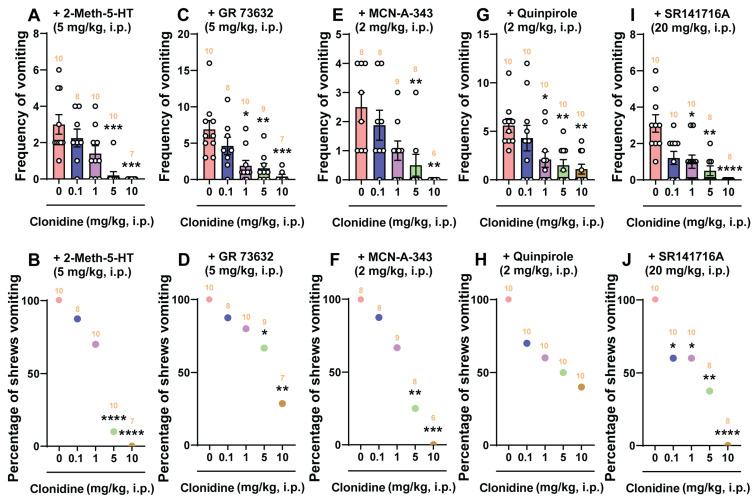
The antiemetic effects of the α_2_-adrenergic receptor agonist clonidine against vomiting evoked by diverse emetogens. Varying doses of clonidine (0, 0.1, 1, 5, and 10 mg/kg, i.p.) were injected to different groups of shrews 30 min prior to an injection of a fully effective emetic dose of selective serotonin 5-HT_3_ receptor agonist 2-Methyl-5-HT (5 mg/kg, i.p., *n* = 7–10 shrews per group) (**A**,**B**), the selective neurokinin NK_1_ receptor agonist GR73632 (5 mg/kg, i.p., *n* = 7–10 shrews per group) (**C**,**D**), the muscarinic M_1_ receptor agonist McN-A-343 (2 mg/kg, i.p., *n* = 6–9 shrews per group) (**E**,**F**), the dopamine D_2/3_ receptor preferring agonist quinpirole (2 mg/kg, i.p., *n* = 10 shrews per group) (**G**,**H**), or the cannabinoid CB_1_ receptor-selective inverse agonist/antagonist SR141716A (20 mg/kg, i.p., *n* = 8–10 shrews per group) (**I**,**J**). Emetic parameters were recorded for the next 30 min. The frequency of emesis was analyzed with Kruskal–Wallis non-parametric one-way ANOVA followed by Dunn’s post hoc test and presented as mean ± SEM (**A**,**C**,**E**,**G**,**I**). The percentage of shrews vomiting was analyzed with chi-square test and presented as the mean (**B**,**D**,**F**,**H**,**J**). * *p* < 0.05, ** *p* < 0.01, *** *p* < 0.001, **** *p* < 0.0001 vs. 0 mg/kg (controls pretreated with vehicle of clonidine). The number of animals in each group is presented on the top of the corresponding column.

**Figure 5 ijms-25-04603-f005:**
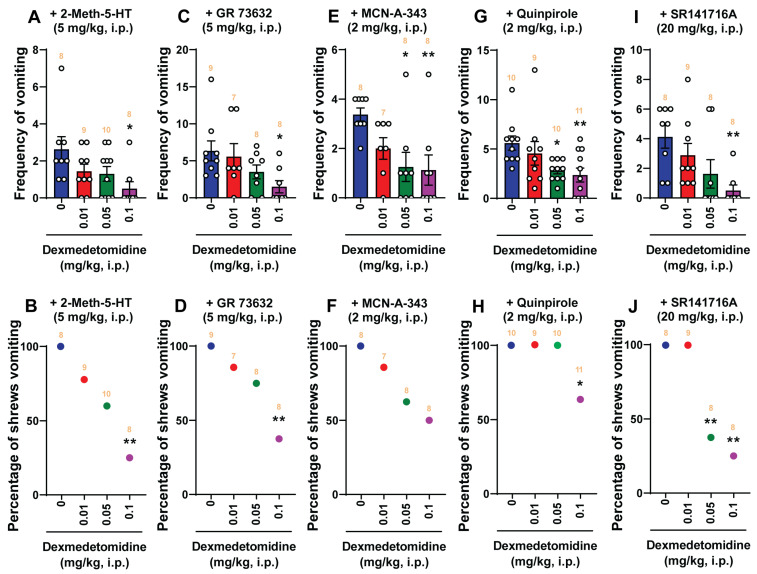
The antiemetic effects of the α_2_-adrenergic receptor agonist dexmedetomidine against vomiting evoked by diverse receptor-selective emetogens. Varying doses of dexmedetomidine (0, 0.01, 0.05, and 0.1 mg/kg, i.p.) were injected to different groups of shrews 30 min prior to an injection of a fully effective emetic dose of the selective serotonin 5-HT_3_ receptor agonist 2-Methyl-5-HT (5 mg/kg, i.p., *n* = 8–10 shrews per group) (**A**,**B**), the selective neurokinin NK_1_ receptor agonist GR73632 (5 mg/kg, i.p., *n* = 7–9 shrews per group) (**C**,**D**), the muscarinic M_1_ receptor agonist McN-A-343 (2 mg/kg, i.p., *n* = 7–8 shrews per group) (**E**,**F**), the dopamine D_2/3_ receptor preferring agonist quinpirole (2 mg/kg, i.p., *n* = 9–11 shrews per group) (**G**,**H**), or the cannabinoid CB_1_ receptor-selective inverse agonist/antagonist SR141716A (20 mg/kg, i.p., *n* = 8–9 shrews per group) (**I**,**J**). Emetic parameters were recorded for the next 30 min. The frequency of emesis was analyzed with Kruskal–Wallis non-parametric one-way ANOVA followed by Dunn’s post hoc test and presented as mean ± SEM (**A**,**C**,**E**,**G**,**I**). The percentage of shrews vomiting was analyzed with chi-square test and presented as the mean (**B**,**D**,**F**,**H**,**J**). * *p* < 0.05, ** *p* < 0.01 vs. 0 mg/kg (controls pretreated with vehicle of dexmedetomidine). The number of animals in each group is presented on the top of the corresponding column.

**Figure 6 ijms-25-04603-f006:**
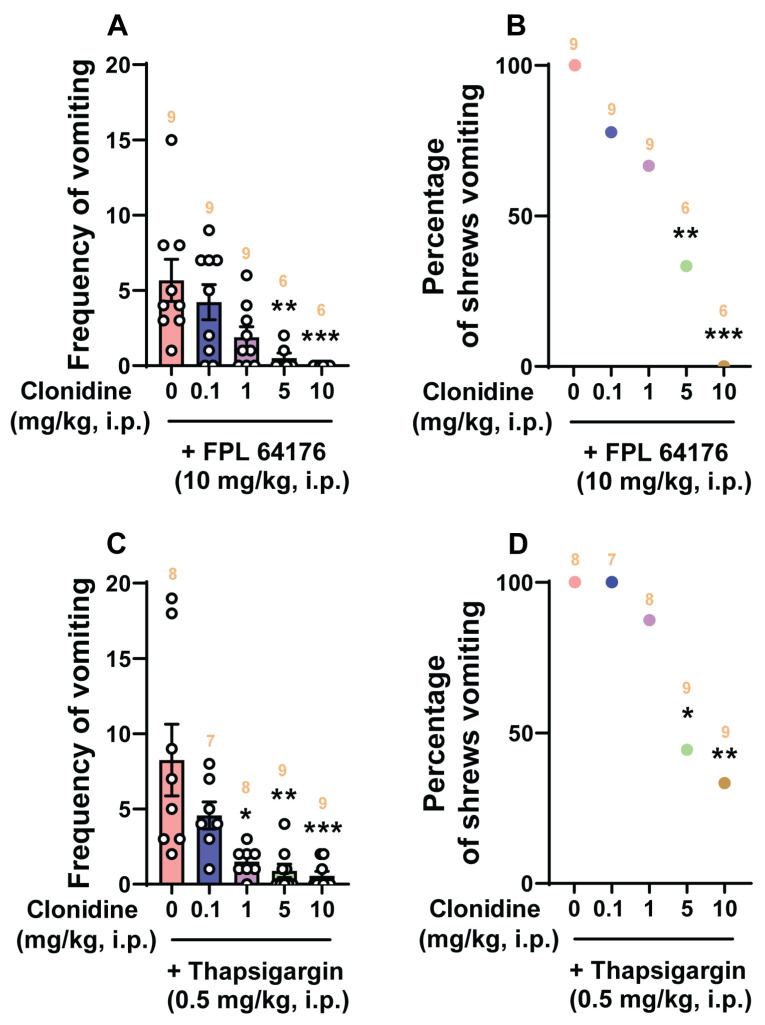
The antiemetic effects of the α_2_-adrenergic receptor agonist clonidine against vomiting caused by Ca^2+^ channel regulators, the LTCC agonist FPL64176 and the SERCA inhibitor thapsigargin. Varying doses of clonidine (0, 0.1, 1, 5, and 10 mg/kg, i.p.) were injected to different groups of shrews 30 min prior to an injection of a fully effective emetic dose of FPL64176 (10 mg/kg, i.p., *n* = 6–9 per group) (**A**,**B**) or thapsigargin (0.5 mg/kg, i.p., *n* = 7–9 per group) (**C**,**D**). Emetic parameters were recorded for the next 30 min. The frequency of emesis was analyzed with Kruskal–Wallis non-parametric one-way ANOVA followed by Dunn’s post hoc test and presented as mean ± SEM (**A**,**C**). The percentage of shrews vomiting was analyzed with chi-square test and presented as the mean (**B**,**D**). * *p* < 0.05, ** *p* < 0.01, *** *p* < 0.001 vs. 0 mg/kg (controls pretreated with vehicle of clonidine). The number of animals in each group is presented on the top of the corresponding column.

**Figure 7 ijms-25-04603-f007:**
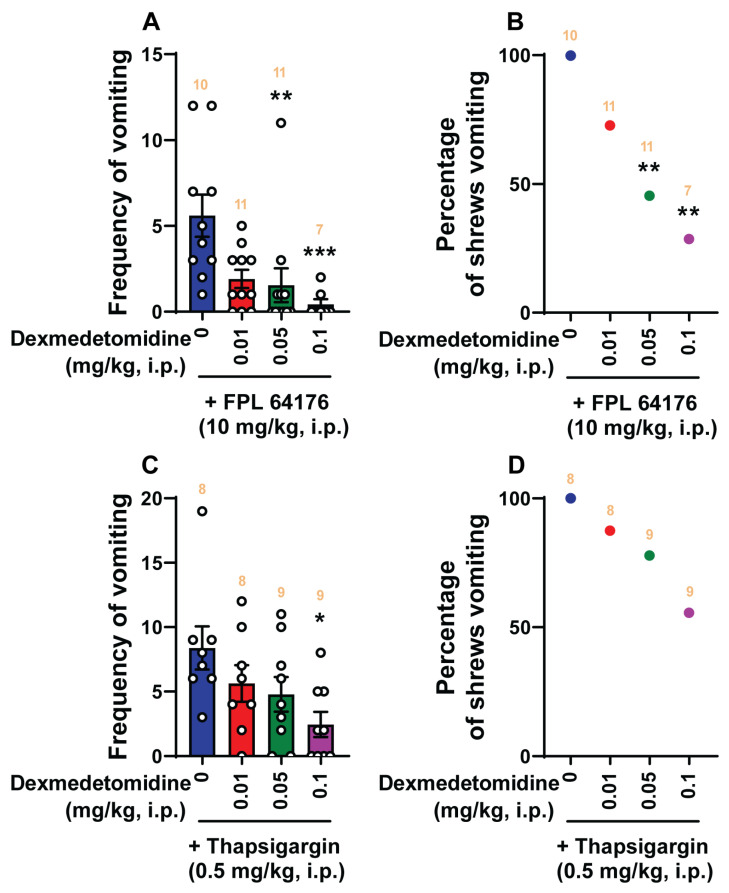
The antiemetic effects of the α_2_-adrenergic receptor agonist dexmedetomidine against vomiting caused by Ca^2+^ channel regulators, the LTCC agonist FPL64176 and the SERCA inhibitor thapsigargin. Varying doses of dexmedetomidine (0, 0.01, 0.05, and 0.1 mg/kg, i.p.) were injected to different groups of shrews 30 min prior to an injection of a fully effective emetic dose of FPL64176 (10 mg/kg, i.p., *n* = 7–11 per group) (**A**,**B**) or thapsigargin (0.5 mg/kg, i.p., *n* = 8–9 per group) (**C**,**D**). Emetic parameters were recorded for the next 30 min. The frequency of emesis was analyzed with Kruskal–Wallis non-parametric one-way ANOVA followed by Dunn’s post hoc test and presented as mean ± SEM (**A**,**C**). The percentage of shrews vomiting was analyzed with chi-square test and presented as the mean (**B**,**D**). * *p* < 0.05, ** *p* < 0.01, *** *p* < 0.001 vs. 0 mg/kg (controls pretreated with vehicle of dexmedetomidine). The number of animals in each group is presented on the top of the corresponding column.

**Figure 8 ijms-25-04603-f008:**
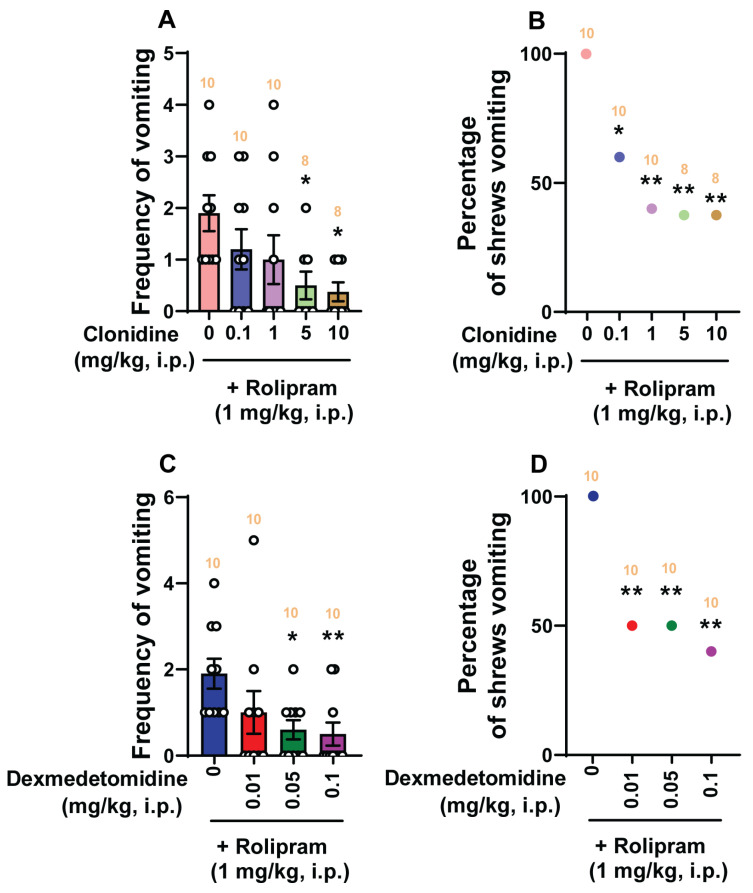
The antiemetic effects of the α_2_-adrenergic receptor agonists clonidine and dexmedetomidine against vomiting caused by the PDE4 inhibitor rolipram (1 mg/kg, i.p.). Varying doses of clonidine (0, 0.1, 1, 5, and 10 mg/kg, i.p., *n* = 8–10 per group; (**A**,**B**)) or dexmedetomidine (0, 0.01, 0.05, and 0.1 mg/kg, i.p., *n* = 10 per group; (**C**,**D**)) were injected to different groups of shrews 30 min prior to an injection of a fully effective emetic dose of PDE4 inhibitor rolipram (1 mg/kg, i.p.). Emetic parameters were recorded for the next 30 min. The frequency of emesis was analyzed with Kruskal–Wallis non-parametric one-way ANOVA followed by Dunn’s post hoc test and presented as mean ± SEM (**A**,**C**). The percentage of shrews vomiting was analyzed with chi-square test and presented as the mean (**B**,**D**). * *p* < 0.05, ** *p* < 0.01 vs. 0 mg/kg (controls pretreated with vehicle of clonidine or dexmedetomidine). The number of animals in each group is presented on the top of the corresponding column.

**Figure 9 ijms-25-04603-f009:**
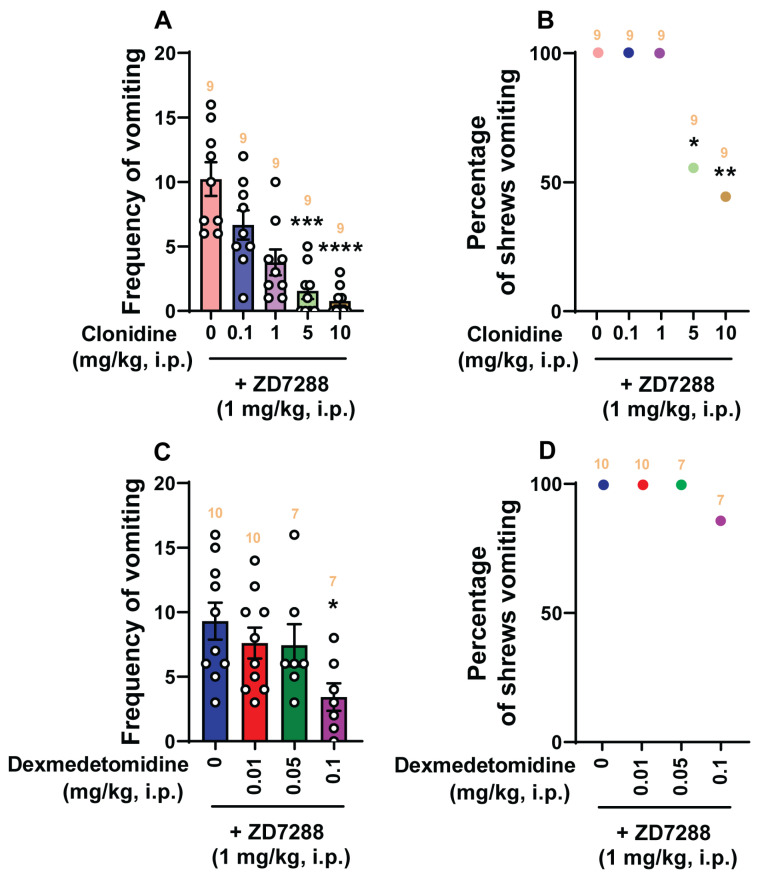
The antiemetic effects of the α_2_-adrenergic receptor agonists clonidine and dexmedetomidine against vomiting caused by the HCN channel blocker ZD7288 (1 mg/kg, i.p.). Varying doses of clonidine (0, 0.1, 1, 5, and 10 mg/kg, i.p., *n* = 9 per group; (**A**,**B**)) or dexmedetomidine (0, 0.01, 0.05, and 0.1 mg/kg, i.p., *n* = 7–10 per group; (**C**,**D**)) were injected to different groups of shrews 30 min prior to an injection of a fully effective emetic dose of ZD7288 (1 mg/kg, i.p.). Emetic parameters were recorded for the next 30 min. The frequency of emesis was analyzed with Kruskal–Wallis non-parametric one-way ANOVA followed by Dunn’s post hoc test and presented as mean ± SEM (**A**,**C**). The percentage of shrews vomiting was analyzed with chi-square test and presented as the mean (**B**,**D**). * *p* < 0.05, ** *p* < 0.01, *** *p* < 0.001, **** *p* < 0.0001 vs. 0 mg/kg (controls pretreated with vehicle of clonidine or dexmedetomidine). The number of animals in each group is presented on the top of the corresponding column.

**Figure 10 ijms-25-04603-f010:**
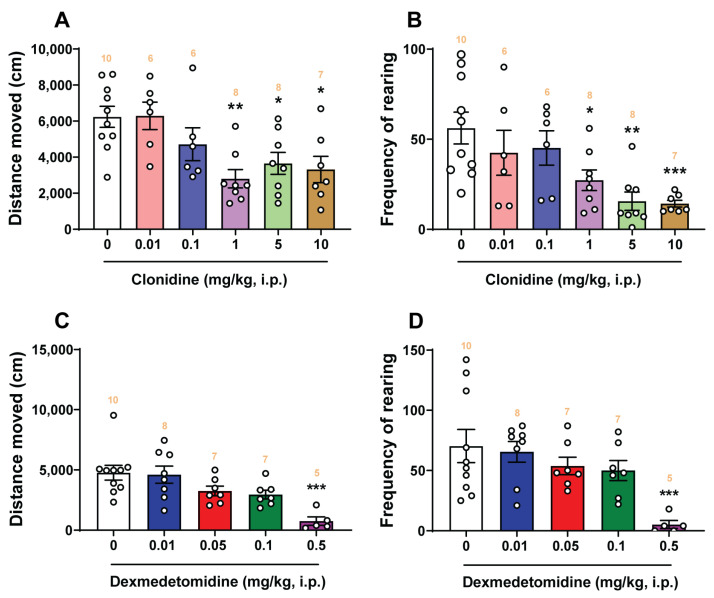
The effect of the α_2_-adrenergic receptor agonists clonidine and dexmedetomidine on open-field locomotor behaviors. Different groups of shrews were injected with either vehicle or varying doses of clonidine (0.1, 1, 5, and 10 mg/kg, i.p., *n* = 6–10; (**A**,**B**))/dexmedetomidine (0.01, 0.05, and 0.1 mg/kg, i.p., *n* = 5–10; (**C**,**D**)) at 0 min. Thirty minutes later, each shrew was individually placed in a white observation cage (27.5 × 27.5 × 28 cm), and the locomotor activity [distance moved (**A**,**C**) and rearing frequency (**B**,**D**)] were recorded for 30 min by a computerized video tracking, motion analysis, and behavior recognition system (Ethovision, version XT 9). The dose-response effects of clonidine and dexmedetomidine on different groups of least shrews were analyzed using the Noldus software. Significant difference relative to the corresponding vehicle group is indicated as * *p* < 0.05, ** *p* < 0.01, *** *p* < 0.001. The number of animals in each group is presented on the top of the corresponding column.

## Data Availability

The raw data supporting the conclusions of this article will be made available by the authors, without undue reservation.

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
