# Peer review of "A Comparative Study of the Antiemetic Effects of α_2_-Adrenergic Receptor Agonists Clonidine and Dexmedetomidine against Diverse Emetogens in the Least Shrew (*Cryptotis parva*) Model of Emesis"

_ijms, 2024, doi:10.3390/ijms25094603_

Round 1
Reviewer 1 Report
Comments and Suggestions for Authors
Sun et al., comparative study of the antiemetic effects of adrenergic α2 receptor agonists clonidine and dexmedetomidine against diverse emetogens in the Least Shrew (Cryptotis parva) model of emesis.Their resultssuggest that dexmedetomidine represents a better candidate for antiemetic potential with advantages over clonidine. This is a well-designed study with solid results. I have several comments for the work.
1.For the experimental design, the sample size calculation and method of randomziation should be presented in the methods section.
2.For the i.p. injections, FPL64176, Thapsigargin and other drugs were dissolved in DMSO/water. Please check if these are correct. In most studies, the saline instead of water for ip injection.
3. For the bar graphs, the authors are encouraged to present the data as bar + scatter dots plot.
4. For the c-fos staining, it is better to use arrows to indicate the positively stained cells in the micrographs.
5.For the 5-HT and SP staining, the image quality is not good enough, as the postive staining is hard to see. Is it possible to increase the magnification to show the results better.
6. Since Yohimbine increased c-fos expression in the brain stem, the authors are recommended to examine the effects of dexmedetomidine and clonidine on the yohimbine-induced c-fos expression in the brainstem.
Author Response
Please see attached file called Response to reviewer 1

Reviewer 2 Report
Comments and Suggestions for Authors
Comments.
Authors investigated whether adrenergic α2 receptor agonists clonidine and dexmedetomidine, or yohimbine (a α2-adrenergic receptor antagonist) evokes vomiting in least shrews (a well-validated experimental emetic model). After, they examined whether the brainstem emetic nuclei mediate the vomitive effect of yohimbine through induction of c-fos, and serotonin- and substance P-release. Additionally, it was investigated whether clonidine and dexmedetomidine have antiemetic effects against several emetogens. Finally, an open-field locomotor analysis. was performed.
The conclusion reached is that clonidine and dexmedetomidine have an antiemetic effect in the least shrew model of yohimbine-induced vomiting and that both α2 receptor agonists have broad-spectrum antiemetic potential, with dexmedetomidine being much more potent without affecting motor behavior.
Observations.
The results describe that yohimbine, clonidine and dexmedetomidine have similar effects both in least shrew and humans, can the authors mention the reason for these similarities as well as the different effects mentioned in other species such as dogs and cats?
Please clarify the reason for using various emetogens. Was it to analyze different pathways of vomiting regulation?
Do all the compounds used in this protocol have clinical applications? If so, it would be illustrative to indicate them.
Authors mention that has been reported that clonidine has an ED50 of 25 μg/kg (i.m.) in dogs and 0.075 μg/kg (i.c.v.) in cats, is the difference in ED50 values due to the species or the route of administration? In relation to this question, the reason for choosing the intraperitoneal route of administration in this protocol must be explained.
Regarding to frequency of vomiting, what are the units of this parameter? It is a bit confusing to read, for example, that yohimbine (1 mg/kg) causes a maximum vomiting frequency of 0.85±0.22.
Indicate or highlight the importance of use of these α2 receptor agonists in the clinic.
Describe in detail the euthanasia procedure.
Define the term ID50 since it is applied in various ways according to the considerations of each study:
https://www.sciencedirect.com/topics/biochemistry-genetics-and-molecular-biology/id50.
https://www.sciencedirect.com/topics/immunology-and-microbiology/id50#:~:text=The%20ID50%20value%20is%20defined,with%20the%20untreated%20virus%20controls.
In addition, it is mentioned the ED50 value of clonidine in cats and dogs, then ED50 and ID50 have the same meaning?
Considering the principle of the 3 Rs for the welfare of animals used for research (Russell and Burch, 1959), the authors indicate that they made efforts to reduce the number of animals used but it would be advisable for them to indicate the total number of animals used in this protocol, how they determined the number of animals for each study group and if there was a need to repeat some experimental procedures.
According to description offered by authors the experimental scheme was the following:
- Control group (treated with vehicle), n = 7 – 14 (for yohimbine treatment), n = 6 – 10 (for clonidine treatment), n = 6 – 10 (for dexmedetomide treatment).
- Yohimbine (0, 0.5, 0.75, 1, 1.5, 2 and 3 mg/kg, i.p.), 7 groups (n = 7 – 14 per group), at least 49 animals were used.
- Clonidine (0, 0.1, 1, 5 and 10 mg/kg, i.p.), 5 groups (n = 6 – 10 per group), at least 30 animals were used (but in legend of figure 1 is indicated that n = 7 – 13 shrews per group, so more animals were used).
- Dexmedetomidine (0, 0.01, 0.05 0.1, 0.5, and 1 mg/kg, i.p.), 6 groups (n = 6 – 10 per group), at least 36 animals were used (but in legend of figure 1 is indicated that n = 8 – 13 shrews per group, so more animals were used).
For drug interaction studies (pro- and anti-emetic evaluation) the experimental scheme was at follows:
- Clonidine (0, 0.1, 1, 5 and 10 mg/kg, i.p.), 5 groups (n = 7 – 13 per group), and 30 minutes after each animal received yohimbine (1 mg/kg, i.p.), at least 35 animals were used.
- Dexmedetomidine (0.01, 0.05, 0.1 mg/kg, i.p.), 4 groups (n = 8 – 13 per group), and 30 minutes after each animal received yohimbine (1 mg/kg, i.p.), at least 32 animals were used. This scheme is not the same as the one described in section 2.1.
For immunohistochemical analysis of c-fos authors indicate that 90 minutes after post systemic administration of yohimbine (1 mg/kg, i.p., n = 6 shrews per group), other 12 animals were used.
Taking advantage of this observation, the following question arises: For the authors, is systemic administration the same as intraperitoneal administration?
For immunohistochemical analysis of 5-HT and SP is described that animals were euthanized at 15- and 30-min post vehicle or yohimbine injection (n = 5 – 6 per group). Then at least other 15 animals were used.
For analysis of antiemetic effects of clonidine (0, 0.1, 1, 5 and 10 mg/kg) against vomiting evoked by the following emetogens:
- 2-Methyl-5-HT (5 mg/kg, i.p., n = 7 – 10 shrews per group),
- GR73632 (5 mg/kg, i.p., n = 7 – 10 shrews per group),
- McN-A-343 (2 mg/kg, i.p., n = 6 – 9 shrews per group),
- Quinpirole (2 mg/kg, i.p., n = 10 shrews per group),
- SR141716A (20 mg/kg, i.p., n = 8 – 10 shrews per group),
- FPL64176 (10 mg/kg, i. p., n = 6 – 9 per group),
- Thapsigargin (0.5 mg/kg, i. p., n = 7 – 9 per group),
- Rolipram (1 mg/kg, i.p.),
- ZD7288 (1 mg/kg, i.p.).
At least 24 animals per each emetic treatment were used.
The same calculation can be done for the case of dexmedetomidine (0, 0.01, 0.05 and 0.1 mg/kg) treatments.
For locomotor behavior analysis the description is the following:
- Clonidine (0.1, 1, 5 and 10 mg/kg, i.p.), n = 6 – 10 per group,
- Dexmedetomidine (0.01, 0.05 and 0.1 mg/kg, i.p.), n = 5 – 10 per group.
It would be convenient to provide an experimental scheme or a detailed description that illustrates the number of animals used in this research.
Russell WMS, Burch RL. The principles of humane experimental technique. Methuen & Co. Ltd.; London, UK: 1959.
Minor observations.
Please:
Correct:
Line 93: the word “recaptor”.
Line 107: there is a hyphen after a square bracket.
Correct the wording of the following phrases:
Lines 113 and 114. “dexmedetomidine (0, 0.01, 0.05 and 0.1, 0.5, 1 mg/kg, i.p., n = 6 – 10 per group)”.
Line 124: “difference was only observed at its 1 mg/kg”.
Sometimes the abbreviation i.p. is written with a space between the letters.
Author Response
Please see to attached file called Response to reviewer 2
